# Risk of SARS-CoV-2 reinfection during multiple Omicron variant waves in the UK general population

Jia Wei [1,2] ✉, Nicole Stoesser [1,3,4,5], Philippa C. Matthews[1,6,7], Tarnjit Khera[8], Owen Gethings[8], Ian Diamond[8], Ruth Studley[8], Nick Taylor[8], Tim E. A. Peto[1,3,4,5], A. Sarah Walker[1,2,4,9,23], Koen B. Pouwels [4,10,23], David W. Eyre [2,3,4,5,23] & the COVID-19 Infection Survey team*

SARS-CoV-2 reinfections increased substantially after Omicron variants emerged. Large-scale community-based comparisons across multiple Omicron waves of reinfection characteristics, risk factors, and protection afforded by previous infection and vaccination, are limited. Here we studied ~45,000 reinfections from the UK's national COVID-19 Infection Survey and quantified the risk of reinfection in multiple waves, including those driven by BA.1, BA.2, BA.4/5, and BQ.1/CH.1.1/XBB.1.5 variants. Reinfections were associated with lower viral load and lower percentages of self-reporting symptoms compared with first infections. Across multiple Omicron waves, estimated protection against reinfection was significantly higher in those previously infected with more recent than earlier variants, even at the same time from previous infection. Estimated protection against Omicron reinfections decreased over time from the most recent infection if this was the previous or penultimate variant (generally within the preceding year). Those 14–180 days after receiving their most recent vaccination had a lower risk of reinfection than those >180 days from their most recent vaccination. Reinfection risk was independently higher in those aged 30–45 years, and with either low or high viral load in their most recent previous infection. Overall, the risk of Omicron reinfection is high, but with lower severity than first infections; both viral evolution and waning immunity are independently associated with reinfection.

Omicron (B.1.1.529) became the dominant severe acute respiratory syndrome coronavirus 2 (SARS-CoV-2) variant globally in December 2021. Due to a high number of mutations in the viral spike protein, it has enhanced transmissibility and infectivity compared with previous variants and can more easily escape immunity acquired from both previous infection and vaccination[1–7], causing widespread infection worldwide. The Omicron variant has further mutated and recombined into different subvariants, which have caused multiple infection waves. In the UK, there were five Omicron infection waves from December 2021 to January 2023, caused by the dominant subvariants BA.1, BA.2, BA.4/5, BQ.1 (a sub-lineage of BA.4/5) and then a mixture of BQ.1 and BA.2.75 and its sub-lineages, particularly CH.1 and XBB.1.5[8].

SARS-CoV-2 reinfections have been reported since mid 2020[9–11]. Before the emergence of the Omicron variant, SARS-CoV-2 reinfections were relatively uncommon[12,13], and previous infections provided 80–90% protection against a pre-Omicron reinfection[14,15]. However, reinfection risk increased substantially with Omicron[15–17], with the protection against an Omicron BA.1 and BA.2 reinfection from a previous non-Omicron infection estimated to be only 20–45%[18–20]. Reinfections with different Omicron subvariants also occur, although

A full list of affiliations appears at the end of the paper. *A list of authors and their affiliations appears at the end of the paper. ✉e-mail: jia.wei@ndm.ox.ac.uk

protection from a previous Omicron infection against a new Omicron reinfection has been found to be greater at 50–90%[18,21]. However, rapid waning of protection against a new BA.5 infection following BA.1/BA.2 infection has also been reported[22].

Understanding the extent and duration of protection against reinfection associated with vaccination and prior infection is important for planning vaccination and other control measures. However, a major challenge is that much information about infection comes from national testing programmes, which may have biases particularly in terms of who decides to test, access to testing (particularly since tests are no longer free for most of the population in many countries, such as the UK), and over-representation of symptomatic infections, leading to incomplete ascertainment of infection history and potential bias in estimating reinfection severity, risk factors, protection from previous infection and vaccination, especially for Omicron infection waves after testing programmes were discontinued, as in the UK from 1 April 2022.

We therefore used data from the UK's Office for National Statistics (ONS) COVID-19 Infection Survey (CIS), which undertook regular longitudinal testing of participants independently of symptoms/ characteristics, linked to data from the English and Welsh national testing programmes and supplemented by non-study positive swab test results self-reported on study questionnaires, to examine the characteristics and severity of confirmed SARS-CoV-2 reinfections compared with first infections. We used flexible parametric survival models to examine the risk factors for reinfections in multiple Omicron waves of the pandemic including those driven by BA.1, BA.2, and BA.4/5 variants, and then a mixture of BQ.1, CH.1, and XBB1.5 subvariants, and specifically investigated the protective effects from previous vaccination and infection.

## Results

We first defined infection episodes using study results from 11,264,965 positive (1.8%) or negative (98.2%) swab tests taken between 26 April 2020 and 13 March 2023 from 535,133 participants (median study participation 24.8 months, IQR 16.4–28.4, range 0–34.5). These study swab tests were taken at approximately monthly intervals (9,189,487 (81.6%) at study worker visits before 31 July 2022 and 2,075,478 (18.4%) from swabs returned by post or courier after the study moved to remote data collection in July 2022, with minimal impact on positivity[23] see Methods), and were supplemented by 254,591 linked positive swab tests from national testing programmes in England and Wales and 197,088 linked self-reported positive swab tests in the same participants (same positive swab could be in both data sources, and could also overlap with study positive swab tests). In total from these data, from 28 February 2020 to 13 March 2023, 245,895 participants ≥18 year were infected with SARS-CoV-2 based on a positive swab test in the study, national testing programmes or self-reported at any time up to their final study assessment (see Methods; self-report only in 43,246 (17.6%) participants). 45,137 (18%) participants were identified as reinfected; 3,513 (1.4%) participants had three confirmed infections, 162 (0.07%) four, and 7 five (Supplementary Table 1, Supplementary Fig. 1).

The median age at first identified infection was 53 years, and 48 and 45 years at the first and second confirmed reinfection (i.e. second and third identified infection), respectively (Table 1). 54.6% of those ever identified as infected were female, versus 58.0% and 64.8% among those identified as reinfected once or twice. 93.4% of those ever identified as infected reported white ethnicity, more than those identified as reinfected once (92.5%) or twice (90.8%). Those reporting working in healthcare were more likely to have reinfections identified, accounting for 5.3%, 6.9%, 9.0%, and 9.3% of those who had one, two, three, and four confirmed infections, versus 4.4% of the cohort as a whole.

38.9% and 39.1% of participants whose first identified infection was with Pre-Alpha or Alpha variants had a second infection identified, with BA.1 and BA.2 being the most common reinfection variant

followed by BA.4/5. 35.6% of participants whose first identified infection was with Delta variant had a second infection, among which 35.1% were BA.4/5 (Supplementary Fig. 1).

Using Kaplan-Meier estimation from the start of the earlier confirmed infection, 50% of participants were identified as reinfected by 799 days (95% Confidence Interval [CI] 786-820) if the earlier infection was with Pre-Alpha variant, versus 733 days (95% CI 719–755) with Alpha variant, and 601 days (591–610) with Delta variant. 25% of participants were reinfected by 521 (516–528), 452 (448–457), 326 (322–331), 368 (364–371), and 390 (382–394) days if the earlier infection was with Pre-Alpha, Alpha, Delta, BA.1, BA.2 variants, respectively (Supplementary Fig. 2).

All of the seven participants identified as having been infected 5 times were white females, aged 21–50 years. Three reported having a long-term health condition, and two were healthcare workers. The variants, symptoms, and Ct values of these infections were shown in Supplementary Table 1.

### Cycle threshold (Ct) values and confirmed reinfections

185,484 (62.9%) infections had a Ct value recorded from the same TaqPath assay as used throughout the study (see Methods). Missing Ct values were almost all either from infections identified from self-report only (52.1% missing Ct; Ct value would not have been known) or national testing programmes only (45.5% missing Ct; either using lateral flow devices (LFDs) or a different PCR test). The median observed Ct was 23 in first, 25 in second, and 26 in third infections ($p < 2.2e-16$) (Table 1).

Using robust linear regression, independently, and allowing for interactions between infection number, variant, and time from vaccination, Ct values were higher in confirmed reinfections than first confirmed infections across different variants, excepting only reinfections with Omicron subvariants in those not vaccinated (Fig. 1). Ct values were lower in males ($p < 2.2e-16$), healthcare workers ($p = 0.002$), those who had a higher deprivation score ($p = 8.0e-10$), those reporting symptoms ($p < 2.2e-16$), and in participants with a higher number of positive tests within confirmed infection episodes ($p = 2.8e-14$) (Supplementary Table 2). Compared with BA.1 variant infections, Pre-Alpha and Alpha variant infections had higher mean Ct values, while Delta had lower mean Ct values ($p = 4.0e-10$). Among different Omicron subvariants, BA.2, BA.4/5, and BQ.1/CH.1.1/XBB.1.5 infections all had progressively higher mean Ct value than BA.1 infections ($p < 2.2e-16$). Being 14–180 days from the most recent vaccination was independently associated with higher Ct values, and there were no substantial differences in this effect across variants and infections, considering interactions. Results remained similar restricting to infections with Ct values measured from the study only (i.e. excluding Ct values from the same Taqpath assay but where this was used in national testing programmes) (Supplementary Fig. 3).

In a separate model only including confirmed reinfections, either lower or higher Ct values in the most recent previous infection were associated with lower Ct values in the current confirmed reinfection ($p = 0.001$) (Supplementary Fig. 4), while reporting symptoms in the most recent previous confirmed infection was associated with higher Ct values in current confirmed reinfection ($p = 0.04$) (Supplementary Table 2).

### Self-reported symptoms and confirmed reinfections

The unadjusted percentage of self-reporting any symptoms was slightly lower in the second (74.0%) and third (75.9%) confirmed infection compared with the first confirmed infection (77.2%) ($p < 2.2e-16$). Among participants with self-reported symptom information from the study (i.e. excluding those with only symptom presence/absence from the national testing programme), the second infection had a slightly lower percentage reporting classic symptoms than the first infection (57.2% vs 58.6%, $p < 2.2e-16$), but there was no

**Table 1 | Characteristics of participants who had a first, second, third, and fourth infection**

|  | First infection (N = 245895) | Second infection (N = 45137) | Third infection (N = 3513) | Fourth infection (N = 162) |
|---|---|---|---|---|
| Age |  |  |  |  |
| Median | 53 | 48 | 45 | 44 |
| IQR | 40, 65 | 38, 59 | 36, 54 | 37, 55 |
| Sex |  |  |  |  |
| Female | 134353 (54.6%) | 26201 (58.0%) | 2276 (64.8%) | 119 (73.5%) |
| Male | 111542 (45.4%) | 18936 (42.0%) | 1237 (35.2%) | 43 (26.5%) |
| Ethnicity |  |  |  |  |
| Non-white | 16329 (6.6%) | 3400 (7.5%) | 323 (9.2%) | 12 (7.4%) |
| White | 229566 (93.4%) | 41737 (92.5%) | 3190 (90.8%) | 150 (92.6%) |
| Reporting working in healthcare |  |  |  |  |
| No | 232772 (94.7%) | 42043 (93.1%) | 3196 (91.0%) | 147 (90.7%) |
| Yes | 13123 (5.3%) | 3094 (6.9%) | 317 (9.0%) | 15 (9.3%) |
| Reporting having a long-term health condition |  |  |  |  |
| No | 185268 (75.3%) | 35247 (78.1%) | 2750 (78.3%) | 116 (71.6%) |
| Yes | 60627 (24.7%) | 9890 (21.9%) | 763 (21.7%) | 46 (28.4%) |
| Deprivation percentile* |  |  |  |  |
| Median | 62 | 60 | 59 | 57 |
| IQR | 38, 82 | 36, 81 | 34, 80 | 29, 77 |
| Source of infection |  |  |  |  |
| CIS only | 49878 (20.2%) | 14743 (32.7%) | 1369 (39.0%) | 61 (37.7%) |
| CIS + national testing/self-reported positive swab test | 84354 (34.4%) | 9973 (22.1%) | 702 (20.0%) | 27 (16.7%) |
| National testing only | 68417 (27.8%) | 7703 (17.1%) | 491 (14.0%) | 30 (18.5%) |
| Self-reported positive swab test only | 43246 (17.6%) | 12718 (28.2%) | 951 (27.0%) | 44 (27.1%) |
| Minimum Ct value in this infection | 157381 (64.0%) | 25948 (57.5%) | 2072 (59.0%) | 83 (51.2%) |
| Median | 23 | 25 | 26 | 24 |
| IQR | 18, 29 | 21, 30 | 22, 31 | 21, 29 |
| Missing (source of missing below) | 88514 (36.0%) | 19189 (42.5%) | 1441 (41.0%) | 79 (48.8%) |
| CIS only | 537 (0.6%) | 310 (1.6%) | 23 (1.6%) | 2 (2.5%) |
| CIS + national testing/self-reported positive swab test | 1391 (1.6%) | 268 (1.4%) | 15 (10.4%) | 3 (3.8%) |
| National testing only | 43246 (48.9%) | 12718 (66.3%) | 951 (66.0%) | 44 (55.7%) |
| Self-reported positive swab test only | 43340 (49.0%) | 5893 (30.7%) | 452 (31.4%) | 30 (38.0%) |
| Report symptoms in this infection |  |  |  |  |
| No symptoms | 56016 (22.8%) | 11739 (26.0%) | 846 (24.1%) | 40 (24.7%) |
| Any symptoms | 189879 (77.2%) | 33398 (74.0%) | 2667 (75.9%) | 122 (75.3%) |
| Classic symptoms (CIS) | 144180 (58.6%) | 25817 (57.2%) | 2062 (58.7%) | 103 (63.6%) |
| Other symptoms only (CIS) | 27108 (11.0%) | 6109 (13.5%) | 553 (15.7%) | 17 (10.5%) |

**Table 1 (continued) | Characteristics of participants who had a first, second, third, and fourth infection**

|  | First infection (N = 245895) | Second infection (N = 45137) | Third infection (N = 3513) | Fourth infection (N = 162) |
|---|---|---|---|---|
| Symptoms from national testing programme | 18591 (7.6%) | 1472 (3.3%) | 52 (1.5%) | 2 (1.2%) |
| Variant of infection |  |  |  |  |
| Pre-Alpha | 11832 (4.8%) | 52 (0.1%) | 0 (0.0%) | 0 (0.0%) |
| Alpha | 16280 (6.6%) | 239 (0.5%) | 0 (0.0%) | 0 (0.0%) |
| Delta | 36749 (14.9%) | 1215 (2.7%) | 11 (0.3%) | 0 (0.0%) |
| Omicron BA.1 | 48867 (19.9%) | 5460 (12.1%) | 121 (3.4%) | 1 (0.6%) |
| Omicron BA.2 | 60839 (24.7%) | 8089 (17.9%) | 345 (9.8%) | 11 (6.8%) |
| Omicron BA.4/5 | 50347 (20.5%) | 15608 (34.6%) | 1227 (34.9%) | 46 (28.4%) |
| Omicron BQ.1/ CH.1.1/XBB.1.5 | 18821 (7.7%) | 14467 (32.1%) | 1809 (51.5%) | 104 (64.2%) |
| Other | 2160 (0.9%) | 7 (0.0%) | 0 (0.0%) | 0 (0.0%) |
| Number of positive swab tests in this infection |  |  |  |  |
| 1 | 164989 (67.1%) | 38536 (85.4%) | 3109 (88.5%) | 134 (82.7%) |
| 2 | 58975 (24.0%) | 5367 (11.9%) | 331 (9.4%) | 20 (12.3%) |
| 3 | 12945 (5.3%) | 752 (1.7%) | 44 (1.3%) | 2 (1.2%) |
| ≥ 4 | 8986 (3.7%) | 482 (1.1%) | 29 (0.8%) | 6 (3.7%) |
| Number of previous vaccinations |  |  |  |  |
| 0 | 32396 (13.2%) | 1286 (2.8%) | 91 (2.6%) | 3 (1.9%) |
| 1 | 6065 (2.5%) | 500 (11.1%) | 29 (8.3%) | 2 (12.3%) |
| 2 | 41373 (16.8%) | 4722 (10.5%) | 347 (9.9%) | 17 (10.5%) |
| 3 | 140073 (57.0%) | 29062 (64.4%) | 2162 (61.5%) | 84 (51.9%) |
| ≥4 | 25988 (10.6%) | 9567 (21.2%) | 884 (25.2%) | 56 (34.6%) |

*A higher deprivation percentile represents less deprived. CIS = COVID-19 Infection Survey. Infections were identified from positive swab tests done within the study, linked from national testing programmes, or self-reported by participants (participants were asked not to self-report study tests). Symptoms were determined by all symptoms from the list below reported by participants within [0,35) days of the first positive test in each infection, including symptoms reported in the last 7 days of study assessments and symptoms reported when participants thought they had COVID. Symptoms were classified as 'classic' (any of cough, fever, loss of taste/smell) or 'other' (myalgia, fatigue/weakness, sore throat, shortness of breath, headache, diarrhoea, nausea, abdominal pain). If no symptoms were reported within CIS, we included any self-reported symptoms available from national testing programmes as another category as specific symptoms were not available from this data source. Variant was defined by sequencing data where available, otherwise by S-gene presence/absence and calendar time reflecting periods when specific variants dominated in the UK (see Methods). In all, 460,026 participants ≥18 years were ever swabbed in the survey, median (IQR) age at first swab was 54 (39–67) years, 53.6% were female, 7.0% reported non-white ethnicity, 4.4% reported working in healthcare and 26.8% a long-term health condition.

evidence of differences with the third and fourth infections. However, the unadjusted percentages reporting only 'other' symptoms (myalgia, fatigue/weakness, sore throat, shortness of breath, headache, diarrhoea, nausea, abdominal pain) were higher in the first and second reinfection than the first infection (13.5%, 15.7% vs 11.0% respectively, $p = 5.7\mathrm{e}{-08}$) (Table 1).

Using a logistic regression model, adjusting for multiple other factors including age, sex, ethnicity, reporting healthcare work, reporting long-term health conditions, social deprivation, time from most recent vaccination, and infection variant, any symptoms were less commonly reported in confirmed reinfections vs first infections (odds ratio OR = 0.64 [95%CI 0.63–0.66] for a first reinfection, and 0.60 [0.55–0.65] for a second reinfection). Results remained similar

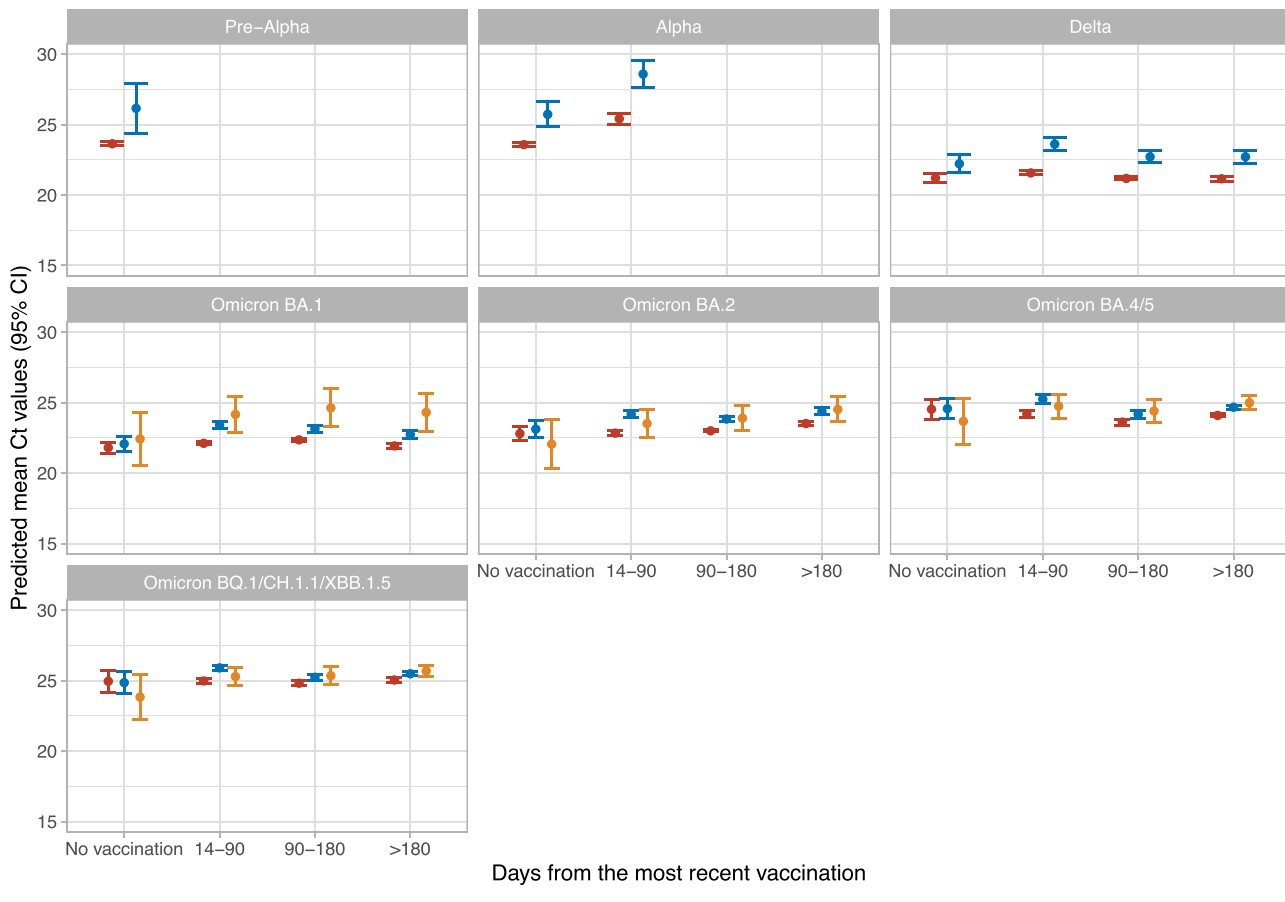

**Fig. 1 | Predicted mean Ct values (95% CIs) by infection variant, days from the most recent vaccination, and infection (first, second, or third).** The 95% CIs are calculated as predictions ± 1.96 × standard error of the predictions. $n = 185{,}484$ infections were included in the model. Predictions are plotted at the reference value of other variables (age = 40 years, female, white ethnicity, not reporting working in healthcare, not reporting having a long-term health condition, deprivation percentile = 60, reporting symptoms, having one positive test in the infection) (estimates shown in Supplementary Table 2).

after additionally adjusting for Ct values of the current infection, with slightly lower ORs (0.59 [0.56–0.60], 0.53 [0.48–0.59] for first and second reinfection vs first infection) (Table 2).

We then fitted a multinomial logistic regression model among participants with self-reported symptoms from study questionnaires only. Classic symptoms were less commonly reported in a first reinfection than a first infection (relative risk ratio RR = 0.60 [95% CI 0.59–0.62]), and were even less commonly reported in a second reinfection than a first infection (RR = 0.54 [0.50–59]). Other symptoms were not significantly different for a first reinfection than a first infection (RR = 0.98 [0.95–1.02]), and even more commonly reported for a second reinfection (RR = 1.20 [1.08–1.34]). Results remained similar for classic symptoms after additionally adjusting for Ct values of the current infection, but other symptoms were less commonly reported in a first reinfection than a first infection after adjusting for Ct (RR = 0.91 [0.87–0.96]) (Table 2).

**The estimated risk of confirmed reinfection in multiple Omicron waves**

42,582, 83,382, 164,263, and 184,566 adults ≥18 year had had a previous confirmed infection and were at risk of reinfection for all or part of time periods where a single variant dominated in the UK, specifically the Omicron BA.1 (27 December 2021 to 6 February 2022), BA.2 (14 March 2022 to 22 May 2022), BA.4/5 (27 June 2022 to 6 November 2022), and BQ.1/CH.1.1/XBB.1.5 (6 November 2022 to 13 March 2023), respectively (denoted "waves"). During these periods, new identified

infections could be most confidently assigned to specific variants, even in the absence of sequencing or S-gene presence/absence (see Methods). Reflecting the widespread vaccine rollout through 2021, 99.5%, 99.8%, 99.9%, and 99.9% participants had received at least one vaccination at the start of each of these respective waves. 3,504, 5,644, 15,079, and 16,076 participants had an Omicron BA.1, BA.2, BA.4/5, or BQ.1/CH.1.1/XBB.1 confirmed reinfection, and the Kaplan-Meier reinfection percentages were 10%, 11%, 14%, and 16% at the end of each wave, respectively, with corresponding reinfection incidences 24, 14, 10, 10 per 10,000 participant days, respectively. The unadjusted confirmed reinfection rate started at very high levels then decreased over time in BA.1 and BA.2 waves, remained lower throughout the BA.4/5 wave, and increased again in BQ.1/CH.1.1/XBB.1 wave, suggesting that the risk was shifting antigenically (Supplementary Fig. 5).

Across waves, using survival models restricted to those with previous confirmed infections (i.e. assessing the risk of confirmed reinfection) with underlying timescale based on calendar date from the start of each wave ($t = 0$), and with time-updated factors for time since most recent vaccination (by number) and most recent prior identified infection (by variant), and adjusting for multiple confounders (see Methods), estimated protection against confirmed reinfection was higher following previous identified infection with more recent variants than earlier variants, including when the time from previous infection overlapped between variants (Fig. 2). In addition, consistently across all Omicron waves, estimated protection against confirmed reinfections decreased over time from the most

**Table 2 | The association (relative risks [RR], 95% confidence intervals (95%CIs)) between reported symptoms and reinfections**

| (a) | No symptom | With symptom | | | |
|---|---|---|---|---|---|
| Model without Ct | | | | | |
| | reference | RR | | 95%CI | *p*-value |
| Infection (2 vs 1) | | 0.64 | | 0.63–0.66 | **<2e–16** |
| Infection (3 vs 1) | | 0.60 | | 0.55–0.65 | **<2e–16** |
| Model adjusted for Ct | | | | | |
| | reference | RR | | 95%CI | *p*-value |
| Infection (2 vs 1) | | 0.59 | | 0.56–0.60 | **<2e–16** |
| Infection (3 vs 1) | | 0.53 | | 0.48–0.59 | **<2e–16** |

| (b) | No symptom | Classic symptoms | | | Other symptoms | | |
|---|---|---|---|---|---|---|---|
| Model without Ct | | | | | | | |
| | reference | RR | 95%CI | *p*-value | RR | 95%CI | *p*-value |
| Infection (2 vs 1) | | 0.60 | 0.59–0.62 | **<2e–16** | 0.98 | 0.95–1.02 | 0.3 |
| Infection (3 vs 1) | | 0.54 | 0.50–0.59 | **<2e–16** | 1.20 | 1.08–1.34 | **0.001** |
| Model adjusted for Ct | | | | | | | |
| | reference | RR | 95%CI | *p*-value | RR | 95%CI | *p*-value |
| Infection (2 vs 1) | | 0.54 | 0.52–0.56 | **<2e–16** | 0.91 | 0.87–0.96 | **0.0002** |
| Infection (3 vs 1) | | 0.50 | 0.45–0.55 | **<2e–16** | 1.12 | 0.98–1.30 | 0.08 |

(a) Multivariable logistic regression model examining any reported symptoms vs no reported symptoms. (b) Multivariable multinomial regression model examining reported classic symptoms (any of cough, fever, loss of taste/smell) and other symptoms (myalgia, fatigue/weakness, sore throat, shortness of breath, headache, diarrhoea, nausea, abdominal pain) among participants with self-reported symptom information from the study (i.e. excluding those with only symptom presence/absence from the national testing programme). Models were adjusted for age, sex, ethnicity, reporting working in healthcare, reporting having a long-term health condition, deprivation percentile, infection variant, time from most recent vaccination. Separate models were built further adjusted for Ct values in the current infection. Two-sided *z*-test was used to test the significance of model coefficients. The 95% CIs are calculated as estimates ±1.96 × standard error of the estimates.

recent identified infection if this was the previous (thick solid lines) or penultimate (thin solid lines) variant (generally reflecting the most recent prior infection being within the preceding year), but did not change or even slightly increased over time if this most recent prior identified infection was with an even earlier variant (dashed lines) (generally more than a year previously) with evidence of statistical heterogeneity between most recent identified infection being with the previous/penultimate vs earlier variant ($p < 0.05$; Tables S3B-E). Thus both the variant of the most recent identified infection and waning of protection from the most recent identified infection if this was with the previous or penultimate variant were independently associated with reinfection risk in most waves. For example, there was no clear evidence that the estimated protection arising from previous Pre-Alpha and Alpha infections changed over time from infection in the BA.1 and BA.2 waves ($p = 0.9$, 0.4 in BA.1, 0.8, 0.9 in BA.2), and somewhat counter-intuitively protection arising from previous Alpha infections increased over time in the BA.4/5 and BQ.1/CH.1.1/XBB.1.5 waves (Alpha: HR = 0.91, 0.85 per 60 days, $p = 0.009$, 0.006, noting that these participants had already avoided reinfection with all variants from Delta onwards). Similarly, estimated protection from a previous Delta infection against a confirmed reinfection in the BA.2 wave gradually declined over time (reinfection HR = 1.14 per 60 days, $p = 4.2e$-05), but did not change over time in the BA.4/5 wave ($p = 0.9$), and increased over time in the BQ.1/CH.1.1/XBB.1.5 wave (HR = 0.89 per 60 days, $p = 1.2e$-07; again these participants had avoided identified reinfection with all previous Omicron variants). Clearer waning of protection over time was seen after Omicron infections, for example, in the BA.4/5 wave, estimated protection decreased over time from previous BA.1 (HR = 1.18 per 60 days, $p = 2.1e$-11) and BA.2 identified infections (HR = 1.32 per 60 days, $p = 8.0e$-08). In BQ.1/CH.1.1/XBB.1.5 wave, the estimated protection was highest after a previous BA.4/5 identified infection, followed by a previous BA.2 identified infection, but estimated protection also waned over time from infection (BA.2: HR = 1.19 per 60 days, $p = 1.7e$-12, BA.4/5: HR = 1.46 per 60 days, $p = 3.7e$-10). Independently, at the same time from the most recent identified previous infection, re-infection in the BQ.1/CH.1.1/XBB.1.5 wave was lower if that most recent infection (within the last 300 days) was BA.4/5 than

BA.2, and BA.2 than BA.1; similarly re-infection in the BA.4/5 wave was lower if that most recent infection (within the last 300 days) was BA.2 than BA.1, and BA.1 than Delta; and re-infection in the BA.2 wave was lower if that most recent infection (within the last 180 days) was BA.1 than Delta. Results remained similar when modelling time from previous identified infection categorically (Supplementary Fig. 6). Results remained similar defining infection episodes also incorporating when participants thought they had had COVID-19, rather than using positive swab tests in the study, linked national testing programmes or as self-reported to try to reduce the number of missed infections (see Methods) (Supplementary Fig. 7).

Time from the most recent vaccination was independently associated with the estimated risk of confirmed reinfection. Compared with being >180 days from the most recent vaccination, the estimated risk of confirmed reinfection was significantly lower 14–90 days and 90–180 days from the most recent vaccination in BA.1, BA.4/5, and BQ.1/CH.1.1/XBB.1.5 waves, but there was no evidence of difference in the BA.2 wave, although estimates were numerically lower (Fig. 3).

Across waves, estimated reinfection risk was consistently higher in young to middle-aged participants (30–45 year) (Supplementary Fig. 8), females, and those reporting white ethnicity, working in healthcare, and long-term health conditions (Supplementary Table 3). Reporting symptoms in the most recent previous identified infection was associated with a lower risk of confirmed reinfection in BA.1, BA.2 waves, but a higher risk in the BQ.1/CH.1.1/XBB.1.5 wave. Participants living in Northern Ireland, Scotland, and Wales had a lower risk of confirmed BA.1 and BA.2 reinfection, but those living in Northern Ireland and Wales had a higher risk of confirmed BQ.1/CH.1.1/XBB.1.5 reinfection (Fig. 4). Estimated reinfection risk was consistently higher in those with an intermediate Ct value (20–30) in previous identified infections across all waves (Supplementary Fig. 9). A 1% higher background prevalence was associated with a 20%, 23%, 33%, and 27% higher risk of confirmed reinfection in the Omicron BA.1, BA.2, BA.4/5, and BQ.1/CH.1.1/XBB.1.5 waves (Supplementary Table 3).

Associations between covariates and the estimated risk of confirmed reinfection remained broadly similar in sensitivity analyses without adjustment for background infection prevalence, with only

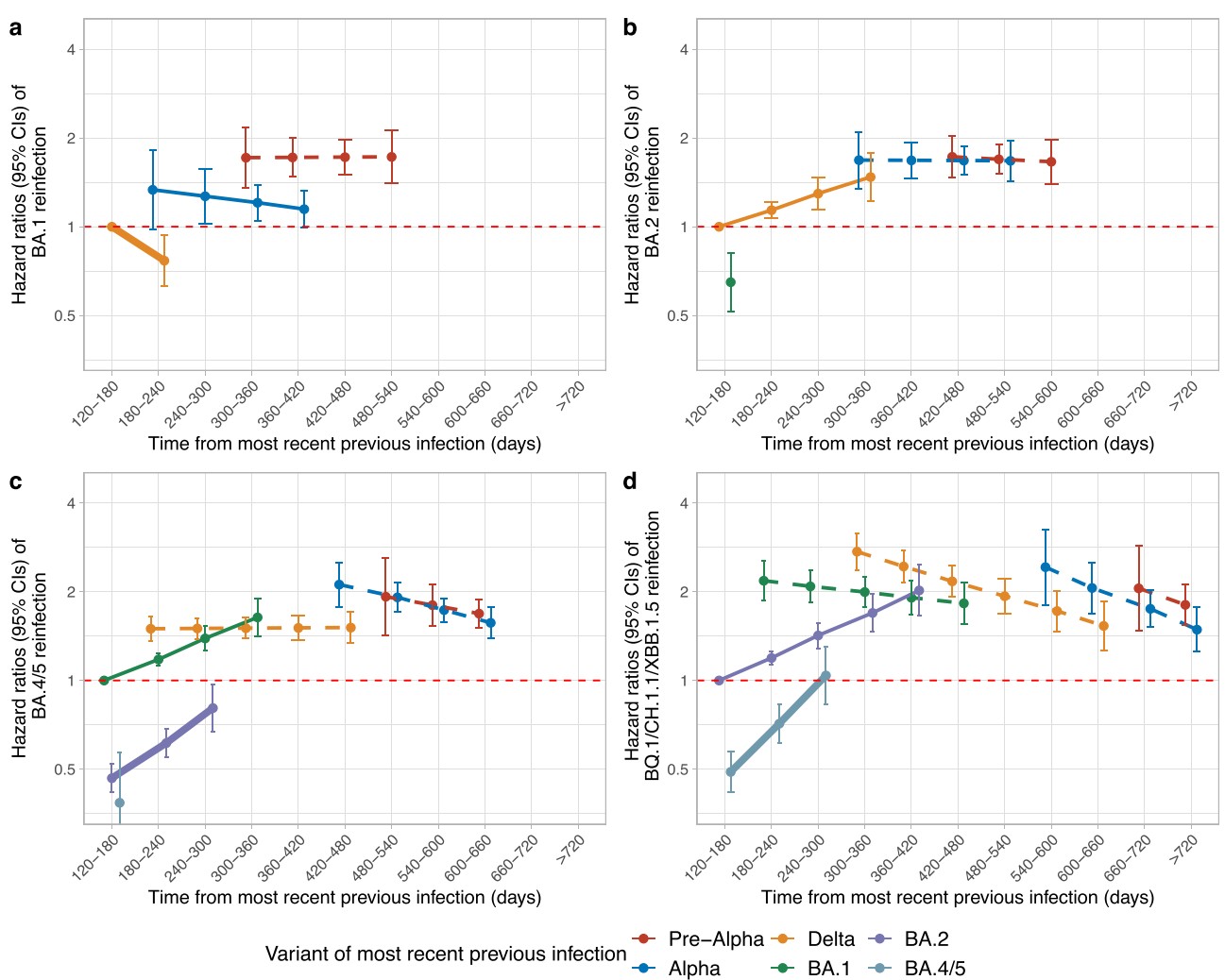

Variant of most recent previous infection
Pre–Alpha · Delta · BA.2
Alpha · BA.1 · BA.4/5

**Fig. 2 | Estimated risk (mean hazard ratios with 95% CIs) of Omicron reinfections in different waves by time from most recent previous infection and variant of most recent previous infection.** *n* = 42,582, 83,382, 164,263, and 184,566 adults who were at risk of Omicron BA.1 (**a**), BA.2 (**b**), BA.4/5 (**c**), and BQ.1/CH.1.1/XBB.1.5 (**d**) reinfections were included in the models, respectively. Time from previous infection was a time-updated covariate categorised as 120–180, 180–240, 240–300, 300–360, 360–420, 420–480, 480–540, 540–600, 600–660, 660–720, >720 days and its effect modelled as a trend over these categories (see Supplementary Fig. 6 for categorical effects). Risk is presented versus a reference category of 120–180 days from an infection in the wave starting ~6 months before the current wave (Delta for BA.1 and BA.2 waves, BA.1 for BA.4/5 waves and BA.2 for BQ.1/CH.1.1/XBB.1.5 waves). Line type and width represent the sequence of variants for better comparisons across waves (thick solid line represents the previous variant, thin solid line represents the penultimate variant, and dashed lines represent earlier variants). The 95% CIs are calculated as exponent of estimates ± 1.96 × standard error of the estimates. Adjusted (Supplementary Table 3) for time-fixed covariates age, sex, ethnicity, reporting working in healthcare, reporting having a long-term health condition, deprivation percentile, infection variant, region, number of previous infections, symptoms in most recent infection whether any previous infection had Ct < 30 or was LFD positive; and time-updated time from most recent vaccination (Fig. 3) and background infection prevalence. Results remain similar in sensitivity analyses without adjustment for background infection prevalence (Supplementary Table 4).

small differences for the effects of age and region (Supplementary Fig. 8, Fig. 4, Supplementary Table 4A). Considering participants as being 'at risk' from the date of their first negative PCR test in the study following each infection rather than 120 days after their previous infection (see Methods) generated similar results; being <120 days from the previous infection was consistently associated with a much lower estimated risk of confirmed reinfection in different waves (Supplementary Table 4B, Supplementary Figs. 10, 11). The associations with sex, ethnicity, and working in healthcare largely disappeared when only including infections defined by positive test results from the study (so not influenced by test seeking behaviour), but estimated protection from previous identified infection remain similar (Supplementary Table 4C).

## Discussion
Here, we have quantified the high confirmed reinfection rates associated with multiple Omicron waves driven by the BA.1, BA.2, BA.4/5,

and then a mixture of BQ.1/CH.1.1/XBB.1.5 variants. Previous studies found the reinfection rate with pre-Omicron variants was <1%[12,24], while others have described reinfection rates with Omicron BA.1/BA.2 of 6%–15%[17,25,26]. In our population, the percentage of those previously identified as infected who were confirmed as reinfected was also around 10-11% at the end of the BA.1 and BA.2 waves, but reached 14–16% at the end of the BA.4/5 and BQ.1/CH.1.1/XBB.1.5 waves, likely reflecting both the longer duration of these waves and viral immune evasion by most recent variants (Fig. 2)[27,28]. We also found that around 1 in 70 participants were identified as having been infected three times, and 1 in 1500 participants four or five times over the 3 year study period, with most fourth or fifth confirmed reinfections in the Omicron BA.4/5 and BQ.1/CH.1.1/XBB.1.5 waves.

Confirmed re-infections were somewhat less likely to be symptomatic, with 40–50% decreased risk of reporting classic symptoms in confirmed reinfections versus first infections; however, one limitation

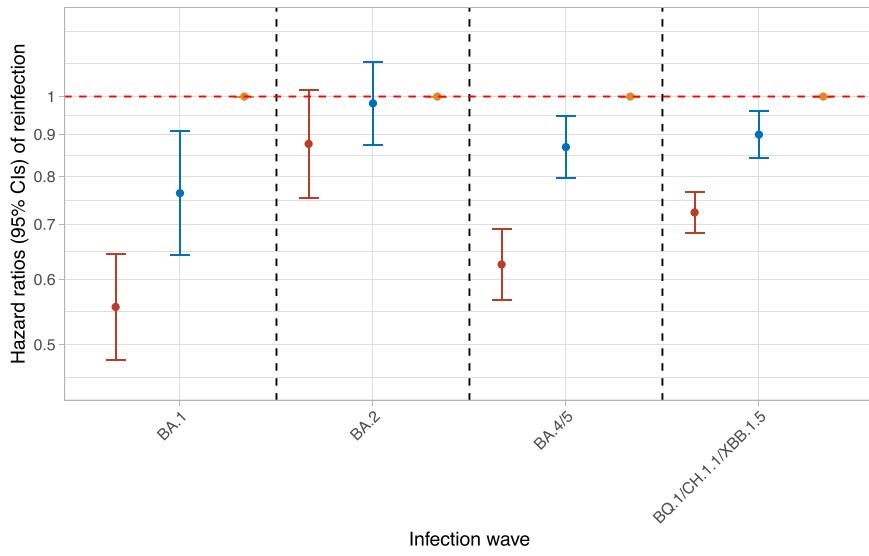

**Fig. 3 | Estimated risk (mean hazard ratios with 95% CIs) of Omicron reinfections in different waves by time from most recent vaccination.** $n$ = 42,582, 83,382, 164,263, and 184,566 adults who were at risk of BA.1, BA.2, BA.4/5, and BQ.1/CH.1.1/XBB.1.5 reinfections were included in the models, respectively. Time from most recent vaccination was a time-updated covariate split into 14–90 days, 90–180 days, >180 days. >180 days was used as reference category. Bivalent vaccines were only used from September 2022 onwards, and therefore would only have been relevant to infections in the final wave. The 95% CIs are calculated by estimates ± 1.96 × standard error of the estimates. Adjusted (Supplementary Table 3) for time-fixed covariates age, sex, ethnicity, reporting working in healthcare, reporting having a long-term health condition, deprivation percentile, infection variant, region, number of previous infections, symptoms in most recent infection, whether any previous infection had Ct < 30 or was LFD positive; and time-updated time from most recent infection (Fig. 2) and background infection prevalence. Results remain similar in sensitivity analyses without adjustment for background infection prevalence (Supplementary Table 4).

is that symptoms were self-reported and inevitably contain an element of subjectivity as well as confounding with other infections. Confirmed reinfections also had higher Ct values, including after adjustment for infection variants, suggesting that identified reinfections were generally less severe, regardless of variants, as Ct values are inversely associated with viral load, which has been associated with the severity and outcomes of SARS-CoV-2 infections[29]. This is consistent with a previous systematic review reporting reinfections were more likely to cause mild illness and were associated with a lower risk of hospitalisation and death compared with the first infections[30]. However, SARS-CoV-2 reinfection may still pose an additional risk of death and hospitalization in people aged >60 years, which may be related to reduced immunity and comorbidities in this group[31]. Therefore, prevention of SARS-CoV-2 reinfection remains particularly important for the older and other high-risk populations.

Using flexible parametric survival models, we estimated the risk of confirmed reinfection over multiple Omicron infection waves (counting time from the start of each wave to account for varying infection pressure over it which could otherwise confound estimates of associations), and specifically examined the protection afforded by previous identified infection and vaccination. Across all four Omicron waves, we found that estimated protection against confirmed reinfection was greatest following previous identified infection with the most recent vs earlier Omicron variants, and with Omicron vs pre-Omicron infections, even at the same time from previous identified infection (Fig. 2). Because variants emerge and calendar time passes together, it is potentially challenging to distinguish whether effects are from changes in variant per se or the time from the most recent infection. However, where the time since the most recent infection overlapped, estimated reinfection risks associated with different previously identified variants showed risk was still generally lower with a more recent variant than an earlier variant at any given time since previously identified infection. Given SARS-CoV-2 has a great ability to evolve and accumulate genetic diversity, these data also indicate that

both viral evolution and the waning of host immunity are independently associated with reinfection.

We found significant waning over time of protection against confirmed BA.4/5 reinfection following previously identified BA.1, BA.2 infections, and against confirmed BQ.1/CH.1.1/XBB.1.5 reinfection following previously identified BA.2 and BA.4/5 infections, consistent with other reports of protection against Omicron reinfection decreasing over time from previous Omicron infections[32]. These analyses used calendar date as the underlying timescale, and estimated the effect of time since most recent infection separately by variant so any remaining 'depletion of susceptibles' bias would mean that the true waning was larger than that observed[33]. We did not find any evidence of waning of protection against a new confirmed Omicron reinfection if the most recent identified infection was with a Pre-Alpha, Alpha, or Delta variant, excepting only previous Delta infections against BA.2 reinfections. However, many of these Pre-Alpha/Alpha/Delta previous identified infections were >360 days previously, particularly for later Omicron waves. Our findings are consistent with waning of protection from previous infection with any variant plateauing after a year, and almost complete immune escape with Omicron BA.1. The estimated risk of confirmed BQ.1/CH.1.1/XBB.1.5 reinfection was similar or even slightly lower 540 days after a most recent identified Pre-Alpha or Alpha infection than that after a most recent identified Delta or BA.1 infection, with similar trends for confirmed BA.4/5 reinfections. This could be explained by a 'healthy survivor effect', as participants with a Pre-Alpha/Alpha infection who remained free from being reinfected in the early three Omicron waves may have better immunity in general and/or have specific behaviours that confer protection (e.g. continued shielding). However, this could also be due to increasing numbers of undetected infections with increasing time from a last prior detected Pre-Alpha/Alpha infection, i.e. increasing 'depletion of susceptibles' in those appearing to still be at risk, meaning that there would most likely be no further change in reinfection risk with time after around 1 year (supported by categorical

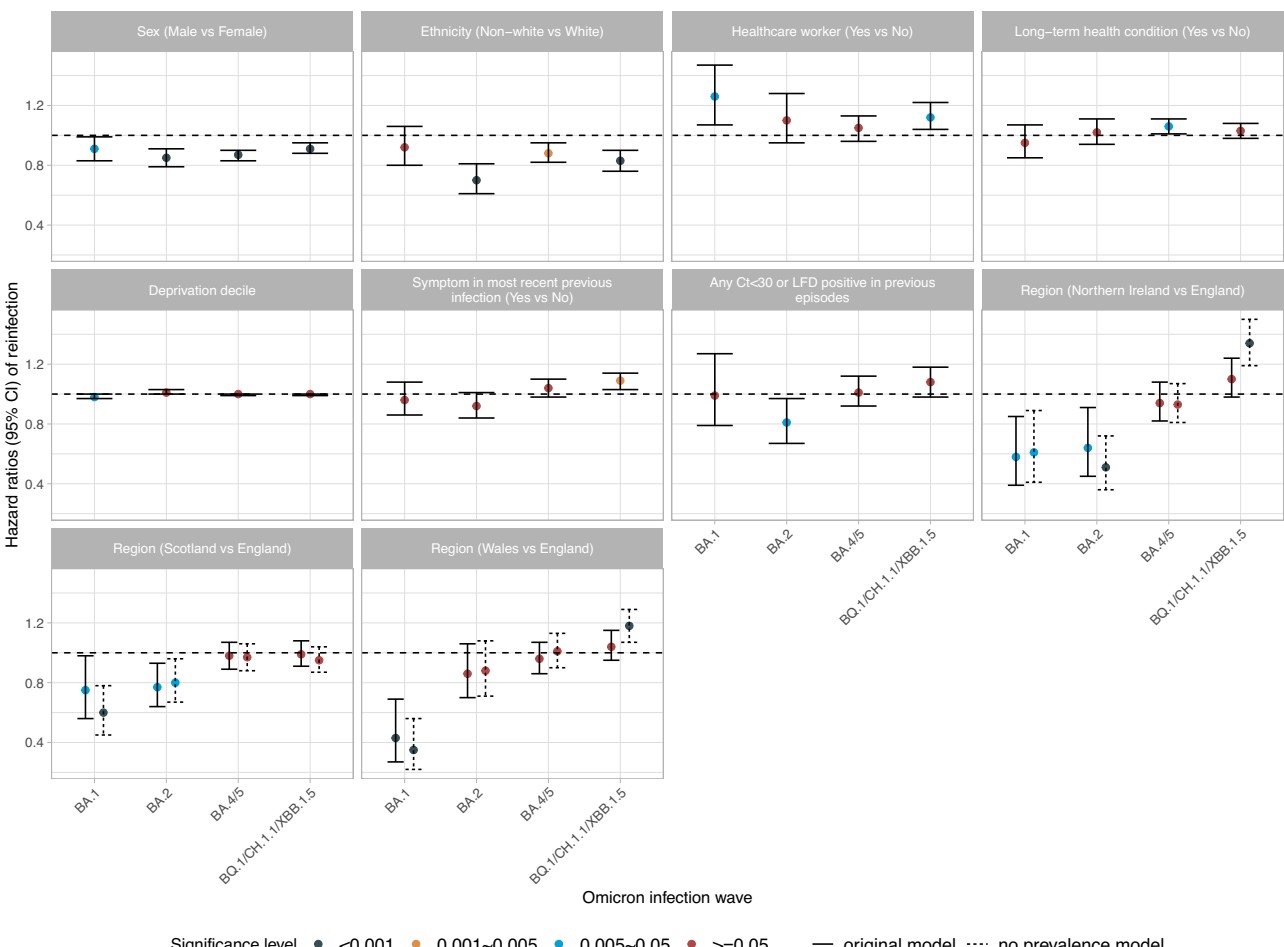

**Fig. 4 | The association (mean hazard ratios with 95%CIs) between different covariates and the estimated risk of reinfection across multiple Omicron infection waves (BA.1, BA.2, BA.4/5, BQ.1/CH.1.1/XBB.1.5).** Estimates shown in Supplementary Table 3. *n* = 42,582, 83,382, 164,263, and 184,566 adults who were at risk of BA.1, BA.2, BA.4/5, and BQ.1/CH.1.1/XBB.1.5 reinfections were included in the models, respectively. Results remain similar in sensitivity analyses without adjustment for background infection prevalence (Supplementary Table 4). Small differences in associations with region between two models are shown in dotted lines.

analysis in Supplementary Fig. 6, although with high uncertainty)[33]. This effect also persisted in sensitivity analyses attempting to reduce the impact of missed infections by including dates when participants thought they had had COVID-19 (but without a positive swab test) in the definition of infection episodes (Supplementary Fig. 7).

We found that the estimated risk of confirmed reinfection increased over time from the most recent vaccination in BA.1, BA.4/5, and BQ.1/CH.1.1/XBB.1.5 waves. Reduced risk of reinfection after vaccination was reported previously for Alpha, Delta, and BA.1 variants[26,34,35], suggesting that among people with previous identified infection, vaccination could provide additional protection against reinfection, although this protection decreased over time. This is consistent with our results that those receiving a vaccination <6 months previously had a lower estimated risk of confirmed reinfection than those >6 months, indicating that people previously identified as infected could still benefit from a more recent vaccination. The lack of evidence for an association in the BA.2 wave could represent lower power (as effect estimates numerically favoured benefit), a true lack of additional benefit from vaccination with this variant, or unmeasured confounding among those clinically vulnerable individuals who should have recently received a fourth dose during the BA.2 wave and would be at a higher risk of reinfection.

We also found that the estimated risk of confirmed reinfection was consistently higher in females, those of white ethnicity, and healthcare workers across different waves, but a sensitivity analysis

restricting infection definitions to be based only on tests done within the study suggested this could mostly be explained by a more frequent test seeking behaviour in these groups. Reporting classic symptoms in the most recent previous identified infection was associated with a lower risk of confirmed BA.1 and BA.2, but not BA.4/5 and BQ.1/CH.1.1/XBB.1.5 reinfections. The estimated risk of confirmed reinfection was much lower in Northern Ireland, Scotland, and Wales than in England during BA.1 and BA.2 waves but was similar during the BA.4/5 and BQ.1/CH.1.1/XBB.1.5 waves. These effects were larger without adjustment for background prevalence, suggesting that they were driven by increased transmission risk overall, rather than being specific to reinfections.

We also found a non-linear U-shaped effect on the estimated risk of confirmed reinfection associated with age and Ct values at the most recent previously identified infection. The risk was higher among those aged 30–45 years, consistent with previous studies on Delta variant[36] and may be explained by working-age adults having greater exposure to social and inter-generational interactions and thus having a higher transmission rate. The lower risk of confirmed reinfection in older participants may be due to both lower social interactions and vaccination policy, as they were the first to receive a third booster vaccination at the end of 2021 and those 75 years and older were also given further booster vaccinations in the first quarter of 2022. The estimated reinfection risk was higher in participants with either a low or high Ct value in their most recent previously identified infection. For those with a very high measured Ct value in their first identified infection,

potentially a relatively less severe initial infection may not generate an effective immune response, leaving these participants more likely to be reinfected. In contrast for those with a very low Ct value, the higher viral load in the first identified infection may indicate a lack of initial viral control that may also be associated with susceptibility to a subsequent infection.

Study strengths include the very large size and unbiased sampling frame of the study, allowing us to unambiguously decompose the relative contribution of different types of previous infection, and time from these previous infections, on reinfection risk. We also used calendar time as our underlying time-scale in time-to-event models, to allow for the fact that background infection pressure changes substantially, even over the course of a wave (Supplementary Fig. 5), and could lead to unmeasured confounding if not adjusted for sufficiently flexibly. We also carefully considered other biases, including immortal time bias.

Study limitations include the fact that our primary analysis chose not to estimate the reinfection risk within 120 days from the most recent previous infection because of challenges identifying short reinfections (requiring sequencing and/or reliable S-gene positivity data, available only for tests with Ct < 30). Our main analyses therefore excluded a number of short reinfections, predominantly with Omicron BA.1 and BA.2. However, results remained very similar in sensitivity analyses including participants from their first negative study PCR test, rather than from 120 days after their previous infection, in order to incorporate shorter reinfections. We did not use a matching approach because the nature of successive infection waves with different variants leads to intrinsic structural confounding between time from most recent previously identified infection and previous variant within each new infection wave (Fig. 2), with similar challenges to target trial emulation approaches. The major limitation is missed infections. The study design was to assess participants every 28–42 days regardless of symptomatology or other participant characteristics, and most intervals between assessments were <45 days (Supplementary Fig. 12a). We also used positive PCR test results from the study, positive test results linked from England and Wales's National Testing programme, and self-reported positive tests from outside the study but reported on study questionnaires to define infection episodes and try to reduce misclassification of both exposure and outcome. However, this means that test seeking behaviour could have an effect on estimates, although this should be minimised by including study results. It is inevitable that we still missed some infections, especially after the emergence of Omicron variants both because of decreased national testing and lower symptomatic fractions, meaning fewer additional positives will have been identified from national testing programmes, but also the slightly shorter duration of PCR positivity, meaning that more infections will have been missed between study tests. However, results remained similar in sensitivity analysis further adding when participants thought they had had COVID-19 (without a positive swab test) into the infection episode definitions to try to reduce missed infections. We did not try use measured spike (S) antibody levels to define prior infection after a participant had been vaccinated, as they were censored at the upper quantification limit of the assay and were repeatedly elevated by further vaccinations. After a very small pilot phase, the survey moved to remote data collection during 11–31 July 2022 after which study worker visits were discontinued and participants returned test kits by post or courier, completing questionnaires online or by telephone. Whilst a minority of participants chose not to continue, the characteristics of those choosing to continue were similar to the original cohort (Supplementary Table 5) and intervals between assessments were also broadly similar (Supplementary Fig.12b). There was minimal impact on positivity comparing swab results during the crossover period 11–31 July 2022[23]. The intervals between assessments were also similar across different waves, making the analyses across waves comparable (Supplementary Fig.12c). Ct

values were missing from 37.1% of identified infections (almost all those self-reported, LFDs or using a different PCR test to the study assay), and Ct values could also be influenced by the timing of infection detection in the study, since tests were performed regardless of symptoms. We could not adjust for laboratory in Ct models as it was not available for tests done outside the survey. However, results remained similar in sensitivity analyses only including Ct values from the survey. We used pre-specified rules to define reinfections based on sequencing data, S-gene presence/absence, Ct values and calendar time (see Methods), but these rules were not perfect, especially for those shorter reinfections, and sequencing could only be attempted on a subset of infections with low Ct. We used calendar time to define the variant of previous infections for those without sequencing data and with only one or no gene-positive, which may be subject to misclassification bias. Symptoms were underreported before the study data collection became remote in July 2022, so the associations between symptoms in reinfections vs first identified infections could be underestimated. We did not have data on hospitalisation and death, thus could not assess disease severity as outcomes.

In conclusion, the estimated risk of confirmed reinfection in the Omicron waves is high, and is associated with both viral evolution and waning immunity, but confirmed reinfections have lower viral loads and fewer symptoms than the first identified infections. A previously identified Omicron infection provides higher estimated protection than a previously identified pre-Omicron infection against new confirmed Omicron reinfections, but this protection decreases over time. Additional protection from vaccination also decreases over time. Given the waning immunity from previous infection and vaccination, and the stronger immune evasion from most recent Omicron variants, reinfection risk remains high, and ongoing evaluation of hybrid immunity against emerging dominant variants remains important to pandemic control and vaccination policy. Further public health measures may be needed to help protect vulnerable people from reinfection.

## Methods

### Population and setting

The ONS COVID-19 Infection Survey (CIS) was a large household survey with longitudinal follow-up (ISRCTN21086382; https://www.ndm.ox.ac.uk/covid-19/covid-19-infection-survey/protocol-and-information-sheets). Private households were randomly selected from address lists and previous surveys on a continuous basis for enrolment from 26 April 2020 through 31 January 2022 (when new recruitment was paused, although follow-up continued until 13 March 2023 when study assessments were paused; 65% were enroled before December 2020, the start of the Alpha wave, and 84% before May 2021, the start of the Delta wave). Following verbal agreement to participate, a study worker visited each selected household to take written informed consent for individuals aged 2 year and over. For those aged 2–15 year, consent was provided by their parents or carers; those 10–15 year also provided written assent. At the first visit, participants were asked for consent for optional follow-up assessments every week for the next month and then monthly subsequently. The study received ethical approval from the South Central Berkshire B Research Ethics Committee (20/SC/0195).

At each assessment, participants were asked about demographics, behaviours (including testing positive on swabs taken outside of the study), work, and vaccination status. Combined nose and throat swabs were taken from all consenting household members for SARS-CoV-2 PCR testing. Blood samples were taken monthly for antibody testing from participants aged 16 year and over in a randomly selected 10–20% of households. Household members of participants who tested positive on a nose and throat swab were also invited to provide blood monthly for follow-up assessments. From April 2021, additional participants were invited to provide blood samples monthly to assess

vaccine responses, based on a combination of random selection and prioritization of those in the study for the longest period (independent of swab test results). Details on the sampling design are provided elsewhere[37]. After 31 July 2022, study worker visits were discontinued and participants could opt-in to continuing to complete questionnaires online or by telephone, returning test kits by post or courier (crossover 11–31 July 2022)[23,38,39]. Data collection was officially paused on 13 March 2023. The distribution of time between study assessments before and after remote data collection, and across different waves were shown in Supplementary Fig.12.

### Vaccination data

Self-reported vaccination data were obtained from participants at assessments, including vaccination type, number of doses, and vaccination dates. Data from participants in England and Wales were also linked to administrative records from the National Immunisation Management Service (NIMS) in England and equivalent in Wales. We used records from administrative data sources where available and otherwise from the study, since linkage was periodic and administrative data sources do not contain information about vaccinations received abroad or in Northern Ireland and Scotland.

### Laboratory testing

Combined nose and throat swabs from the CIS were analysed at the UK's national Lighthouse Laboratories at Milton Keynes and Glasgow using identical methodology. PCR for three SARS-CoV-2 genes (N protein, S protein, and ORF1ab) was performed using the Thermo Fisher TaqPath RT-PCR COVID-19 kit, and analysed using UgenTec FastFinder 3.300.5, with an assay-specific algorithm and decision mechanism that allows conversion of amplification assay raw data from the ABI 7500 Fast into test results with minimal manual intervention. Positive samples are defined as having at least a single N and/or ORF1ab gene detected, and PCR traces exhibited an appropriate morphology. The S gene alone is not considered to be positive[37]. These test results had both gene positivity and cycle threshold (Ct) values available. During periods of high sample returns, a small number of CIS swabs were tested at two other laboratories, one using an endpoint PCR test (no Ct or gene positivity data available). The Ct values from Milton Keynes and Glasgow laboratories were compared by ONS accounting for the stage of the epidemic and there was no evidence of differences.

For participants in England and Wales, we also included positive swab test results (SARS-CoV-2 PCR and lateral flow device tests) linked from national clinical/hospital-based testing (also including health and care workers) and community testing programmes[40]. We did not have the exact laboratory information for these tests, but a substantial proportion of these additional tests were performed at the Lighthouse Laboratories (two above plus one further at Liverpool) using the same Taqpath PCR test as used in CIS: we used gene positivity and Ct values from this Taqpath PCR test but not other tests or laboratories in the national testing programme. As these linked positive test results were not available for participants from Scotland and Northern Ireland, we also included self-reported positive swab tests from study questionnaires (no Ct or test type available). In sensitivity analyses, we additionally included dates when participants reported thinking that they had had COVID-19 (414,096 such reports), but without confirmation from a positive swab test. This potentially might further reduce the number of missed infections, particularly from early pre-Alpha, Alpha and Delta infection waves when testing was not available or before some participants had joined the study, but could increase misclassification bias, thus we did not include this information in the main analyses.

### SARS-CoV-2 reinfection definition

We first grouped repeated positive tests from the sources above into infection episodes, which were then used in all analyses. To reflect the fact that some individuals test positive on PCR for extended periods of time when testing is independent of symptoms/case contacts as in this study (in contrast to national testing programmes), whereas others have reinfections (confirmed by sequencing) after only short periods of time, we incorporated information from genetic sequencing, S-gene presence/absence, and cycle threshold (Ct) values, together with negative PCR test results from CIS only. Using criteria developed through expert consensus, careful inspection and analysis of CIS data alone, and considering definitions of reinfection used elsewhere[15,31,41,42], we defined the start of a new infection episode as any of the following: (1) a new swab positive occurring >120 days after an index positive with the preceding test being negative based on analyses of vaccine effectiveness against Delta and Alpha variants which showed that definitions based on shorter periods of time and/or without a previous negative in these earlier calendar periods erroneously included those testing PCR-positive for long periods of time[43], or (2) >90 days with the preceding two consecutive tests being negative (one negative after 20 December 2021 when Omicron variants dominated given higher re-infection rates with Omicron[15,17]), or (3) >60 days with the three preceding consecutive tests being negative, or (4) after 4 preceding consecutive negative test results at any time.

We then split these infection episodes if they had grouped together positive tests containing multiple sequences from different genetic lineages (e.g. BA.5 and BA.2), or had incompatible S-gene target positivity consistent with co-circulating variants with Ct < 30 (e.g. S-gene positive and S-gene negative, both with Ct < 30 during periods when BA.1 and BA.2 were co-circulating), or had large decreases in Ct or low Ct long after the first positive within an episode (both indicative of a new infection rather than ongoing PCR positivity). We also split infection episodes where a new lateral flow device positive was recorded 19 days or more after the first positive in an infection episode, since this again indicates high viral load and actively replicating virus, more likely associated with a new infection.

### SARS-CoV-2 reinfection variants, Ct values and symptoms

The variant associated with each identified infection episode was determined by whole genome sequencing where this was available, otherwise as Pre-Alpha/Delta/Omicron BA.2-compatible if the S-gene was detected (with N/ORF1ab/both), or as Alpha/Omicron BA.1/Omicron BA.4/5-compatible if positive at least once for ORF1ab+N (but not for the S gene, S-gene target failure, SGTF), using the dates when these variants were dominant in the UK, i.e. accounted for >50% of infections. For those without sequencing data and with only one gene-positive (N-only/ORF1ab-only) or no gene positivity/Ct data available, we assigned them to each variant type based on the national dominant circulating variant in each surveillance week (>50% of positive tests in the study): Pre-Alpha (before 06 December 2020), Alpha (07 December 2020 to 16 May 2021), Delta (17 May 2021 to 12 December 2021), Omicron BA.1 (13 December 2021 to 20 February 2022), Omicron BA.2 (21 February 2022 to 5 June 2022), and Omicron BA.4/5 infections (6 June 2022 to 6 November 2022). After 7 November 2022, sequencing data in the UK showed a mixture of different sub-lineages including the BQ.1 variant (a sub-lineage of BA.5), CH.1.1 variant (a sub-lineage of BA.2.75), and XBB variant (a recombinant lineage derived from two BA.2 sub-lineages), so this wave was denoted 'BQ.1/CH.1.1/XBB.1.5'.

The Ct value associated with each identified infection was the minimum observed across all positive tests within each episode performed using the same Taqpath PCR test at the Lighthouse laboratories, whether this test was done within the study or at these laboratories by the national testing programme, taking the mean Ct value across all detected targets per test (and then the minimum across tests).

Associated symptoms were defined as all symptoms from the list below reported by participants on study questionnaires within [0,35]

days of the first positive test in each infection episode[44], including symptoms reported in the last 7 days at assessments and symptoms reported at dates when participants reported they had had COVID. These symptoms were classified as 'classic' (any of cough, fever, loss of taste/smell) or 'other' (any of another 8 symptoms solicited consistently since the start of the study (myalgia, fatigue/weakness, sore throat, shortness of breath, headache, diarrhoea, nausea, abdominal pain))[45]. If no symptoms were reported within CIS, we also included any self-reported symptoms available from routine national testing as "other" as specific symptoms were not available from these data sources.

## Statistical analysis

We included participants ≥18 year at their first identified infection episode who had at least one identified infection episode from 28 February 2020 (first positive swab test in linked or self-reported data) to 13 March 2023 (end of study data collection). Data from all eligible participants were included, but age was truncated at 85 years for those >85 years to reduce the influence of outliers (0.7% of all participants).

We first investigated how Taqpath Ct values (a surrogate for viral load and potentially infection severity; lower Ct values indicate higher viral loads[29]) varied by infection episode (initial infection, first reinfection, second reinfection) using a robust linear regression model, adjusting for the variant of the current infection, age, sex, ethnicity, reporting working in patient-facing healthcare, report having a long-term health condition, deprivation percentile, self-reported symptoms (classic, other) in the current infection episode, number of positive tests in an infection episode, and time from the most recent vaccination (no vaccination, <14 days, 14–90 days, 90–180 days, and >180 days). Pairwise interactions between the infection episode, variant, and time from the most recent vaccination were included to examine the relationships between these different factors and Ct values as these were highly significant. A separate robust linear regression model estimated associations between the Ct value of confirmed reinfections (second and third infections) and Ct value and self-reported symptoms in the most recent previous identified infection. Other covariates remained the same. Ct values in the previous identified infection were fitted using restricted cubic splines with 3 knots to account for nonlinearity.

We then examined how the percentage of confirmed infections where participants reported symptoms (no symptom, classic symptoms, other symptoms) varied by infection episode (initial infection, first reinfection, second reinfection) using a multinomial regression model, adjusting for the same covariates as above: variant of the current infection episode, age, sex, ethnicity, reporting working in patient-facing healthcare, report having a long-term health condition, deprivation percentile, Ct values in the current infection episode, and time from the most recent vaccination (no vaccination, <14 days, 14–90 days, 90–180 days, and >180 days). Interactions were not included because there was no evidence suggesting that including the interaction terms improved model fit.

We used time-to-event survival models to estimate the risk of confirmed reinfection, treating calendar date as the underlying time scale and using flexible parametric survival models (stpm2)[46,47] to reflect the substantial changes in background infection rate with each wave, modelling the baseline log-cumulative hazard using B-splines. The optimal number of knots for the splines was selected based on minimizing the Akaike information criterion (AIC). Separate survival models were built for reinfections that were identified during each variant wave (defined above), using the date of start of the wave as time 0. To reduce misclassification bias, we restricted the time periods modelled to those surveillance weeks when a given variant accounted for ≥85% of infections based on S-gene positivity: Omicron BA.1 (27 December 2021 to 6 February 2022), Omicron BA.2 (14 March 2022–22 May 2022), Omicron BA.4/5 (27 June 2022–6 November

2022), and Omicron BQ.1/CH.1.1/XBB.1.5 subvariants (7 November 2022–13 March 2023; this final period had both S-gene positive and negative variants circulating at <85% and >15%). Counting participants as being "at risk" from their previous index positive date could cause 'immortal time bias'[48] because reinfections could not happen until the participant stopped testing positive repeatedly. Whilst in theory subsequent positive tests could be assigned to a new infection episode on the basis of sequencing, S-gene presence/absence and/or Ct values as described above, as sequencing was only performed and S-gene presence/absence can only be reasonably reliably detected in samples with low Ct, potential for this immortal time bias remains. Therefore, we defined an 'at-risk date' following each infection as the first date a participant could have been counted as having reinfection had they tested positive based only on the sliding scale of time and number of previous negative tests. Each participant entered the risk set at the later of the start of the wave (time 0) or the date they were first at-risk of reinfection using a late-entry method. They were censored at the last date of the wave, or the last known date of their study assessments, whichever was earlier. Thus analyses included previous infections only identified through non-study tests, but not infections identified after study participation stopped. 67% of participants with any infection were recruited before 7 December 2020, the start of the Alpha wave.

We then excluded time at risk <120 days from previous infection to minimise immortal time bias from varying durations of observed PCR positivity, thus ensuring that risk sets were not very small and unrepresentative shortly after previous infection. This excluded a number of observed reinfections from each wave (603 (17%) Omicron BA.1, 1194 (21%) Omicron BA.2, 720 (5%) Omicron BA.4/5, 348 (2%) Omicron BQ.1/CH.1.1/XBB.1.5). For sensitivity analyses, we counted participants as being 'at risk' from the date of their first negative study PCR test (i.e. not excluding time at risk <120 days from a previous infection and thus including shorter reinfections).

We used a multivariable model to examine associations between risk of confirmed reinfection and the following time-fixed covariates; continuous age (16–85 years), sex, ethnicity (white vs. non-white due to small numbers), reporting having a long-term health condition, reporting working in patient-facing healthcare, deprivation percentile, geographical region (England, Scotland, Wales, Northern Ireland), Ct values in the most recent previously identified infection, symptoms in the most recent previous infection (classic and other symptoms), had any Ct < 30 or lateral flow device (LFD) positive in previous infections, and the number of previous infections. Age and Ct values were fitted using restricted cubic splines with 3 knots to account for non-linear effects. To examine waning immunity from natural infection or vaccination, we further added two time-updated covariates: time from previous infection (split into 120–180, 180–240, 240–300, 300–360, 360–420, 420–480, 480–540, 540–600, 600–660, 660–720, >720 days), and time from most recent vaccination (no vaccination, 14–90 days, 90–180 days, >180 days after the most recent vaccination). Time from previously identified infection was modelled categorically and as a trend across these categories in separate models. We did not assess whether the effect of these time-fixed and time-updated covariates changed over each infection wave, given their relatively short duration (2–4 months), limiting power.

We further adjusted for infection prevalence in the survival models to account for the influence from the background infection pressure. In this way, we could try to separate the risk of being exposed to the virus from those being reinfected given exposure. A generalised additive model (GAM) was built using all study swab PCR test results as the denominator and positive study PCR results as the outcome. Calendar time and age were included using a tensor product spline which was allowed to vary by region/country. Predicted values from the GAM model were used to represent the prevalence varied by calendar time, age, and region, which were then joined to the dataset and included as a covariate in the survival models. Separate models

without adjustment for background infection prevalence were fitted as sensitivity analyses.

Analyses were performed in R 4.0 using the following packages: tidyverse (version 1.3.1), ggsankey (version 0.0), MASS (version 7.3-53), nnet (version 7.3–12), arsenal (version 3.4.0), gmodels (version 2.18.1), ggeffects(version 0.14.3), cowplot (version 1.1.1), emmeans (version 1.5.1), survival (version 3.2–7), survminer(version 0.4.8), and rstpm2 (version 1.5.2).

## Reporting summary
Further information on research design is available in the Nature Portfolio Reporting Summary linked to this article.

## Data availability
De-identified study data are available for access by accredited researchers in the ONS Secure Research Service (SRS) for accredited research purposes under part 5, chapter 5 of the Digital Economy Act 2017. Individuals can apply to be an accredited researcher using the short form on https://researchaccreditationservice.ons.gov.uk/ons/ONS_registration.ofml. Accreditation requires completion of a short free course on accessing the SRS. To request access to data in the SRS, researchers must submit a research project application for accreditation in the Research Accreditation Service (RAS). Research project applications are considered by the project team and the Research Accreditation Panel (RAP) established by the UK Statistics Authority at regular meetings. Project application example guidance and an exemplar of a research project application are available. A complete record of accredited researchers and their projects is published on the UK Statistics Authority website to ensure transparency of access to research data. For further information about accreditation, contact Research.Support@ons.gov.uk or visit the SRS website.

## Code availability
A copy of the analysis code is available at https://github.com/jiaweioxford/COVID19_reinfection. https://doi.org/10.5281/zenodo.10436334.

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

## Acknowledgements

We are grateful for the support of all COVID-19 Infection Survey participants. This study is funded by the UK Health Security Agency and the Department of Health and Social Care with in-kind support from the Welsh Government, the Department of Health on behalf of the Northern Ireland Government and the Scottish Government. J.W. is supported by University of Oxford and the China Scholarship Council. A.S.W., T.E.A.P., N.S., K.B.P. are supported by the National Institute for Health Research Health Protection Research Unit (NIHR HPRU) in Healthcare Associated Infections and Antimicrobial Resistance at the University of Oxford in partnership with the UK Health Security Agency (UK HSA) (NIHR200915). A.S.W. and T.E.A.P. are also supported by the NIHR Oxford Biomedical Research Centre. K.B.P. is also supported by the Huo Family Foundation. A.S.W. is also supported by core support from the Medical Research Council UK to the MRC Clinical Trials Unit [MC_UU_12023/22] and is an NIHR Senior Investigator. P.C.M. is funded by Wellcome (intermediate fellowship, grant ref 110110/Z/15/Z) and holds an NIHR Oxford BRC Senior Fellowship award. DWE is supported by a Robertson Fellowship and an NIHR Oxford BRC Senior Fellowship. NS is an Oxford Martin Fellow and holds an NIHR Oxford BRC Senior Fellowship. The views expressed are those of the authors and not necessarily those of the National Health Service, NIHR, Department of Health, or UKHSA. For the purpose of Open Access, the author has applied a CC BY public copyright licence to any Author Accepted Manuscript version arising from this submission. This work contains statistical data from ONS which is Crown Copyright. The use of the ONS statistical data in this work does not imply the endorsement of the ONS in relation to the interpretation or analysis of the statistical data. This work uses research datasets which may not exactly reproduce National Statistics aggregates.

## Author contributions

The study was designed and planned by A.S.W., I.D. and K.B.P. and is being conducted by A.S.W., T.K., O.G., R.S., N.T., T.E.A.P., P.C.M., N.S., D.W.E. and the COVID-19 Infection Survey Team. This specific analysis was designed by J.W., D.W.E., A.S.W. and K.B.P. J.W. contributed to the statistical analysis of the survey data. J.W., D.W.E., K.B.P. and A.S.W. drafted the manuscript and all authors contributed to interpretation of the data and results and revised the manuscript. D.W.E., K.B.P. and A.S.W. contributed equally. All authors approved the final version of the manuscript.

## Competing interests

DWE declares lecture fees from Gilead, outside the submitted work. PCM receives GSK funding to support a PhD fellowship in her team. All other authors declare no competing interests.

## Additional information

[1]Nuffield Department of Medicine, University of Oxford, Oxford, UK. [2]Big Data Institute, Nuffield Department of Population Health, University of Oxford, Oxford, UK. [3]Department of Infectious Diseases and Microbiology, Oxford University Hospitals NHS Foundation Trust, John Radcliffe Hospital, Oxford, UK. [4]The National Institute for Health Research Health Protection Research Unit in Healthcare Associated Infections and Antimicrobial Resistance at the University of Oxford, Oxford, UK. [5]The National Institute for Health Research Oxford Biomedical Research Centre, University of Oxford, Oxford, UK. [6]The Francis Crick Institute, 1 Midland Road, London, UK. [7]Division of infection and immunity, University College London, London, UK. [8]Office for National Statistics, Newport, UK. [9]MRC Clinical Trials Unit at UCL, UCL, London, UK. [10]Health Economics Research Centre, Nuffield Department of Population Health, University of Oxford, Oxford, UK. [23]These authors contributed equally: A. Sarah Walker, Koen B. Pouwels, and David W. Eyre. ✉e-mail: jia.wei@ndm.ox.ac.uk

## the COVID-19 Infection Survey team

Emma Rourke[9], Tina Thomas[9], Dawid Pienaar[9], Joy Preece[9], Sarah Crofts[9], Lina Lloyd[9], Michelle Bowen[9], Daniel Ayoubkhani[9], Russell Black[9], Antonio Felton[9], Megan Crees[9], Joel Jones[9], Esther Sutherland[9], Derrick W. Crook[1], Emma Pritchard[1], Karina-Doris Vihta[1], Alison Howarth[1], Brian D. Marsden[1], Kevin K. Chau[1], Lucas Martins Ferreira[1], Wanwisa Dejnirattisai[1], Juthathip Mongkolsapaya[1], Sarah Hoosdally[1], Richard Cornall[1], David I. Stuart[1], Gavin Screaton[1], Katrina Lythgoe[2], David Bonsall[2], Tanya Golubchik[2], Helen Fryer[2], John N. Newton[11], John I. Bell[12], Stuart Cox[13], Kevin Paddon[13], Tim James[13], Thomas House[14], Julie Robotham[15], Paul Birrell[15], Helena Jordan[16], Tim Sheppard[16], Graham Athey[16], Dan Moody[16], Leigh Curry[16], Pamela Brereton[16], Ian Jarvis[17], Anna Godsmark[17], George Morris[17], Bobby Mallick[17], Phil Eeles[17], Jodie Hay[18], Harper VanSteenhouse[18], Jessica Lee[19], Sean White[20], Tim Evans[20], Lisa Bloemberg[20], Katie Allison[21], Anouska Pandya[21], Sophie Davis[21], David I. Conway[22], Margaret MacLeod[22] & Chris Cunningham[22]

[11]Office for Health Improvement and Disparities, London, UK. [12]Office of the Regius Professor of Medicine, University of Oxford, Oxford, UK. [13]Oxford University Hospitals NHS Foundation Trust, Oxford, UK. [14]University of Manchester, Manchester, UK. [15]UK Health Security Agency, London, UK. [16]IQVIA, London, UK. [17]National Biocentre, Milton Keynes, UK. [18]Glasgow Lighthouse Laboratory, London, UK. [19]Department of Health and Social Care, London, UK. [20]Welsh Government, Cardiff, UK. [21]Scottish Government, Edinburgh, UK. [22]Public Health Scotland, Edinburgh, UK.

