## [Peer Review File · Nature Communications]

Risk of SARS-CoV-2 reinfection during multiple Omicron variant waves in the UK general populationREVIEWER COMMENTS

Reviewer #1 (Remarks to the Author):

This article addresses an important question of the extent and durability of protection against SARS-CoV-2 conferred by previous infection and vaccination in the Omicron period including recently emerged sub variants such as XBB. The study uses an extremely large dataset of systematically tested individuals from UK surveillance. Main results include high rates of reinfection, higher levels of protection from more recent variants, substantial waning but also waning-independent changes in reinfection risk associated with the emergence of new variants. The paper includes a great deal more useful information on the effect of prior infection in Ct values and symptoms which could be helpful in projecting future epidemic waves and furthering our understanding of the effects of immunity as the virus becomes endemic. The study is well conducted and presented.

I have a few comments.

Major comment. While the study includes individuals systematically tested on a monthly basis there still is a risk of missed infections, likely increasing in Omicron period because of decreased routine testing and lower symptomatic fractions. While this is discussed in the limitations it doesn't seem to be really addressed in sensitivity analyses. In the methods it seemed that for a subset of individuals there were paired serology at each visit. I was wondering whether these data could be analysed to provide estimates of the numbers of missed infections over time and these could then be used to explore how missed infections could have biased estimates. In particular the unexpected finding in some cases of protection increasing with increasing time since vaccination could to me be well explained by missed infections which then conferred protection.

The authors present a sensitivity analysis which accounts for community force of infection. Given that this was likely a real and important contribution to observed effects I wonder why this was not the main analysis and could this be swapped around.

Minor comments

Reviewer #2 (Remarks to the Author):

This manuscript describes the result of a prospective cohort study for SARS-CoV-2 reinfection conducted among 245,895 adults in the UK's national COVID-19 Infection Survey with at least one infection.

The evidence provided addresses important, interesting, and poorly studied questions. Notable strengths in this regard are the population-level data, follow up of study participants, and the focus on reinfections during Omicron. However, weaknesses of this paper are the methodology and reporting of results. Testing data and study methods and in the main manuscript should be more thoroughly described.

In this review I have highlighted a few concerns about the design, analysis, and interpretation.

1. Study design:

Understanding protection from previous infections is an important causal question that can greatly benefit from high quality observational data. Conclusions would be strengthened by using causal inference methods that can more directly address statistical issues such as selection bias and balance.

One example of this is with questions around waning protection. If you have two groups with differing levels of protection, the number of susceptible individuals will decrease more rapidly in the group with less protection. Over time, this could look like waning protection (i.e. protection seems less effective), but in reality, it could simply be a result of selection bias. Lipsitch et al describes this phenomenon as it relates to vaccine waning (cited below).

Lipsitch M, Goldstein E, Ray GT, Fireman B. Depletion-of-susceptibles bias in influenza vaccine waning studies: how to ensure robust results. *Epidemiol Infect* (2019);147:e306.

Accounting for the depletion-of-susceptibles bias is especially important with longer follow-up times, as is common in most waning studies. One potential way to address this issue of selection bias is to use a matched study, censoring matched pairs at the same time, to reduce any arm from depletion-of-susceptibles, as seen in Dagan et al (cited below).

Dagan N, Barda N, Kepten E, et al. BNT162b2 mRNA Covid-19 Vaccine in a Nationwide Mass Vaccination Setting. *N Engl J Med* (2021);384(15):1412–23.

The authors do mention “healthy-survivor bias” in the discussion; however, I would have liked to have seen more attempts to control for biases discussed in the limitations within the study design, especially as they compare HR between models.

2. Hazard ratios:

The authors study an ambitious number of questions, including Ct values, waning protection, and protection against variants. However, there seems to be strong assumptions around variables of interest. Hazard ratios were calculated per 60 days after their most recent infection by a variant, though if the goal is to study hazard ratios over time (as it’s presented in Figure 2), it is unclear why time-varying HR were not used. Hernán provides a discussion of some of these issues (cited below).

Hernán MA. The Hazards of Hazard Ratios. *Epidemiology* (2010);21(1):13–5.

3. Testing:

Testing is one of the most important factors in understanding bias as it relates to detection of infections. There should be more discussion of biases because of testing, and how authors accounted for this bias in the study design. The authors state that:

"From 28 February 2020 to 13 March 2023, 245,895 participants $\geq 18y$ were infected with SARS-CoV-2 based on a positive swab test in the study, national testing programmes or self-reported at any time up to their final study assessment (see Methods; self-report only in 43,246 (17.6%))."

However, the percentage of self-reports (17.6%) is for first infection only, rising to 28.2% for their first reinfection. The CIS study also stopped in-person visits 14 July 2022, and collected opt-in questionnaires and testing (mailed). Does this mean that routine testing only occurred between enrollment and 14 July 2022? Would tests that were mailed in count as CIS or self-report? What proportion of participants mailed in their tests, and did they do it on-schedule? Even prior to 14 July 2022, did study workers successfully visit all participants on-schedule, and if not, what was the missingness? Since enrollment continued until 31 January 2022, when a substantial proportion of the population had already been infected, what assumptions were made regarding previous infections? How did the study design account for some of these biases? These questions are all critical to understand potential biases in data collection. Depending on the answers to these questions, it may also make analyses for most of BA.4/5 wave, and all of BQ.1/CH.1.1/XBB.1.5 wave distinct, and potentially not comparable with other waves.

4. Ct values:

The Ct analysis provides interesting information, but their discussion should be done cautiously and include caveats of issues raised around interpreting Ct values (more below). Because testing was not frequent (mostly monthly) and not always routine, there may be a number of reasons why Ct values increased with reinfections, such as shortened duration of reinfections, less vigilant self-testing among those with previous infections, etc.

It may also be advisable to filter the Ct analysis to only include specimens analyzed in the same laboratory and using the same method. There was also a large fraction of specimens with missing

Ct values, and there should be a greater discussion of how the missing values occurred, and whether they were systematic in any way as to bias results.

According to a joint statement by the Infectious Diseases Society of America and Association for Molecular Pathology,

"Ct values may not be comparable for individual patients tested sequentially with the same method and are definitely not directly comparable across different real-time PCR tests or testing laboratories."

IDSA and AMP joint statement on the use of SARS-CoV-2 PCR cycle threshold (Ct) values for clinical decision-making. March 12, 2021. <https://www.idsociety.org/globalassets/idsa/public-health/covid-19/idsa-amp-statement.pdf>

Minor comments:

1. The authors use strong causal language, and more measured language is recommended. For example,

"Protection" -> "Estimated protection"

"Risk against" -> "Estimated risk"

"Reinfection" -> "Detected/confirmed reinfections"

2. The background prevalence model seems to be more conservative and account for additional sources of variation. Is there a reason why it isn't the primary model?

3. In the introduction, the authors claim that routine testing programmes were a limitation of prior studies, but routine testing is known to reduce detection bias.

4. Page 12:

"Although the study design was to assess participants every 28-42 days regardless of symptomatology, and most intervals between assessments were <45 days".

Should this be >45 days?

Reviewer #3 (Remarks to the Author):

This manuscript presents a sweeping, yet in-depth look at reinfections in the UK during the various Omicron waves through early 2023. It finds that reinfections are common, with Kaplan-Meier estimates over 50% of those infected with earlier variants being reinfected during the Omicron waves. Other key findings are that infection with a recent variant seems to offer significant protection against reinfection for a time, though infection with earlier variants offer much less protection. Likewise, vaccination offers protection against reinfection that wanes over time. Reinfections appeared to generally be less significant disease, with higher Ct values and fewer symptoms.

While the study design precludes any estimate of the absolute reduction in infection risk from a prior infection, it does provide an important look at how that protection changes over time and changes in variant, as well as the impact of vaccination and patient characteristics on risk.

The methods are sound and multiple sensitivity looks at the primary analysis (tables S3-4, figures 2, S7-9) give confidence in the results.

The in-depth look at the seven patients with five infections feels like a case series report inserted into the larger study.

I believe these data are the least important (or at least most tangential to the main thrust of the paper), and yet it is presented in the first paragraph of the results. It could be removed or moved later in the results.

Symptom data are incomplete, with symptoms identified through national testing classified as "other" even though the specific symptoms are unknown. These symptoms could be either classic or other, so I believe that it's inappropriate to classify them as other. These unknown symptoms

should be treated as a separate category or excluded, or symptoms could be treated as yes/no for the primary analysis.

The differing definitions of reinfection when treating it as an exposure or an outcome adds complexity and makes the methods harder to follow and the results less generalizable to other studies with less-complex definitions. The logic of each definition seems sound, but I wonder if a single, simpler definition for both wouldn't be more appropriate.

Similarly, the authors use a stricter definition when assigning variants as an outcome than as an exposure. How many events were excluded by this stricter definition, and did the rate of exclusion vary by variant? This information could help inform on how much potential there is for misclassification bias.

30 September 2023

Dear Reviewers,

Thank you for your critical assessment of our study, responding to which has significantly improved our manuscript. Please find our point-by-point responses to the comments below.

Best wishes,

Jia Wei, Sarah Walker, Koen Pouwels, David Eyre, on behalf of all co-authors.

Reviewer #1 (Remarks to the Author):

This article addresses an important question of the extent and durability of protection against SARS-CoV-2 conferred by previous infection and vaccination in the Omicron period including recently emerged sub variants such as XBB. The study uses an extremely large dataset of systematically tested individuals from UK surveillance. Main results include high rates of reinfection, higher levels of protection from more recent variants, substantial waning but also waning-independent changes in reinfection risk associated with the emergence of new variants. The paper includes a great deal more useful information on the effect of prior infection in Ct values and symptoms which could be helpful in projecting future epidemic waves and furthering our understanding of the effects of immunity as the virus becomes endemic. The study is well conducted and presented.

I have a few comments.

Major comment. While the study includes individuals systematically tested on a monthly basis there still is a risk of missed infections, likely increasing in Omicron period because of decreased routine testing and lower symptomatic fractions. While this is discussed in the limitations it doesn't seem to be really addressed in sensitivity analyses. In the methods it seemed that for a subset of individuals there were paired serology at each visit. I was wondering whether these data could be analysed to provide estimates of the numbers of missed infections over time and these could then be used to explore how missed infections could have biased estimates. In particular the unexpected finding in some cases of protection increasing with increasing time since vaccination could to me be well explained by missed infections which then conferred protection.

Response: We acknowledge that there is a risk of missed infections even with regular testing from the design of the study, independently of symptoms or participant characteristics. This was the reason that we had already included positive swab test results from England and Wales's National Testing programme, and self-reported positive tests, in our definition of infection episodes in order to try to reduce misclassification of both exposure and outcome, as described in the original methods.

We did have paired serology data on spike (S) antibody for a subset of individuals throughout the study and for nucleocapsid (N) antibody for a shorter period of time (discontinued due to funding constraints after vaccine rollout). However, it is not

straightforward to use S antibody data to define prior infection after vaccination, as repeated vaccinations increased S antibody levels to different degrees in different individuals, followed by different degrees of waning, as did infections, and over the course of the study three different dilutions were used in the assay as overall antibody levels rose, leading to measured S antibody levels being censored at different upper limits of quantification in different time periods. We are currently investigating dynamic time warping machine learning methods to try to use anti-spike and anti-nucleocapsid antibody data to identify infections and then unlink vaccination from S-antibody changes, but this is complex and not currently precise enough to integrate into this study, as well as being available only on a subset of participants. We have added this to the limitation, although we would note that it is difficult to see how missed infections would be preferentially in those further from their last prior Alpha/pre-Alpha infection in the BA.4/5 and later waves (note that the reviewer was incorrect – we found increasing protection – or rather lower risk of reinfection - with increasing time from previous *infection* (Figure 2), not increasing time from vaccination (Figure 3)).

However, to examine the robustness of our results, we further performed a sensitivity analysis by additionally including in our definition of infection episodes dates when participants reported thinking that they had had COVID-19, but without confirmation from a positive swab test. This added more infections to the models, including pre-Alpha, and thus could potentially reduce the number of missed infections, but results remain very similar to our main models, and the lower risk of reinfection associated with increasingly distant previous infections still exists (as shown below). We did not use this as the main model because self-reporting 'thinking that I have had COVID-19' without confirmed testing could introduce more bias to the classification of infection, but have added these as sensitivity analyses to the methods and results (Figure S7).

Nevertheless, reporting thinking that one has had COVID-19 would still be subject to the limitations the reviewer mentioned, e.g. lower symptomatic fractions in the Omicron period. However, we would like to emphasise that our study design, incorporating test results from the abovementioned sources, would miss many fewer infections than other similar studies examining reinfections using national testing data.

The authors present a sensitivity analysis which accounts for community force of infection. Given that this was likely a real and important contribution to observed effects I wonders why this was not the main analysis and could this be swapped around.

Response: We did not use the models adjusted for community infection rate as the main analyses in the beginning because we estimated the infection rate using study swab PCR test results adjusted for time, age, and region; thus this was only an approximation to the true infection prevalence. The results were very similar in models with and without adjustment for prevalence, especially associations with the time from the most recent infection, and our conclusions were unaffected. Therefore we originally used the models without adjustment for prevalence as the main analyses to estimate the overall risk of reinfection as an outcome, and the models adjusted for prevalence as sensitivity analyses to separate the risk of being exposed to the virus and the risk of being reinfected given exposure.

However, given the reviewer’s preference (and also comments from reviewer 2 below), we have swapped this so that the models adjusted for background prevalence are now the main analyses and the original models are sensitivity analyses. We have modified all the figures and tables accordingly throughout the manuscript.

Reviewer #2 (Remarks to the Author):

This manuscript describes the result of a prospective cohort study for SARS-CoV-2 reinfection conducted among 245,895 adults in the UK's national COVID-19 Infection Survey with at least one infection.

The evidence provided addresses important, interesting, and poorly studied questions. Notable strengths in this regard are the population-level data, follow up of study participants, and the focus on reinfections during Omicron. However, weaknesses of this paper are the methodology and reporting of results. Testing data and study methods and in the main manuscript should be more thoroughly described.

In this review I have highlighted a few concerns about the design, analysis, and interpretation.

1. Study design:

Understanding protection from previous infections is an important causal question that can greatly benefit from high quality observational data. Conclusions would be strengthened by using causal inference methods that can more directly address statistical issues such as selection bias and balance.

One example of this is with questions around waning protection. If you have two groups with differing levels of protection, the number of susceptible individuals will decrease more rapidly in the group with less protection. Over time, this could look like waning protection (i.e. protection seems less effective), but in reality, it could simply be a result of selection bias. Lipsitch et al describes this phenomenon as it relates to vaccine waning (cited below).

Lipsitch M, Goldstein E, Ray GT, Fireman B. Depletion-of-susceptibles bias in influenza vaccine waning studies: how to ensure robust results. *Epidemiol Infect* (2019);147:e306.

Accounting for the depletion-of-susceptibles bias is especially important with longer follow-up times, as is common in most waning studies. One potential way to address this issue of selection bias is to use a matched study, censoring matched pairs at the same time, to reduce any arm from depletion-of-susceptibles, as seen in Dagan et al (cited below).

Dagan N, Barda N, Kepten E, et al. BNT162b2 mRNA Covid-19 Vaccine in a Nationwide Mass Vaccination Setting. *N Engl J Med* (2021);384(15):1412–23.

The authors do mention “healthy-survivor bias” in the discussion; however, I would have liked to have seen more attempts to control for biases discussed in the limitations within the study design, especially as they compare HR between models.

Response: These are very important points and we address 'depletion of susceptibles' and 'target trial emulation' approaches in turn. We would however stress that we consider that their application to assessing the impact of previous infection with different variants

is extremely complex, and would welcome any further thoughts from the reviewer as exactly how to balance the different trade-offs involved.

First, in terms of depletion of susceptibles, we would note that we believe that we have taken exactly the approach suggested by Lipsitch et al (2019) in that our survival models for new infections in the BA.1, BA.2, BA.4/5 and BQ.1/CH.1.1/XBB waves only include those with previous infections (equivalent to those previously vaccinated in that paper), condition on calendar date (our survival models use calendar date as the underlying timescale) and use time-updated covariates for time since previous infection. We have clarified this in the Results section (was, and still remains, in the original Methods). We have also clarified that the vast majority of our population was vaccinated at the start of these intervals, as we do present estimates of vaccine waning, although this is not the primary focus of this paper.

Further, the "depletion of susceptible bias" (as defined in Lipsitch et al, although technically a bias due to an excess of individuals not at risk/at reduced risk in the risk-set due to exposure misclassification as a consequence of missed recent infections and therefore could potentially be better defined as enrichment of non-susceptibles bias) occurs when there are unobserved infections in the population that occur differentially between two groups being compared, leading to biased comparisons because one group has lower observed risk than true risk, suggesting waning of protection when there is not. Not conditioning on calendar time is one cause of such a bias when comparing time from previous infection/vaccination, but we did this, as clarified in Results. Another key driver could be test seeking behaviour, for example if one group (eg those unvaccinated) systematically tested more often; but here we rely on the design of the study which included regular testing in all participants, and we only included follow-up in survival models when participants were regularly testing within the study. We did supplement this with information about testing from national testing programmes, so accept that some bias from differential missed infections could still remain, although we do feel this is less plausible between groups defined by variant of previous infection at the same time from previous infection, than between eg unvaccinated vs vaccinated individuals.

Further, as per our response to Reviewer 1 above, we have added a sensitivity analysis including information on dates when participants thought they had COVID-19 without a positive swab test into our definitions of infection episodes, to try to reduce the impact of missed infections (whilst noting that this may also affect misclassification by including episodes which were not truly COVID-19). Results are very similar (Figure S7) (added to Results and Discussion).

However, we do agree that "depletion of susceptibles" could be another reason to 'healthy survivor bias' in explaining the lower estimated risk with increasing time from distant prior Pre-Alpha/Alpha infection, since the further from these previous infections, the more likely it is that an intervening infection could have been missed. (We note that this is the opposite direction to the waning effect which the reviewer was initially concerned about: following Table 1 in Lipsitch 2019, our estimates of waning shortly after previous infection should in fact be under-estimates). This would mean that the true effect at longer times since the last previous infection was flat, ie once the previous infection was

a year previously there was no further change in risk (supported to some extent by the categorical analysis in Figure S6, although 95% CI are wide here). We have added this point to the Discussion with the Lipsitch 2019 reference. However, we do not think there are further analyses we can do to adjust for this “anti-waning” (the opposite direction to the reviewer’s original concern above).

Second, in terms of the “target trial” emulation approaches as used by Dagan et al, we note that there is some controversy as to whether these should be used with essentially non-randomisable interventions such as “infection with variant X” vs “infection with variant Y” (originally in the context of estimating “causal” effects of co-infection with TB in HIV-infected individuals), in contrast to vaccination vs no vaccination which is a randomisable intervention. In the causal inference literature this is called the consistency assumption, i.e. the need for a well-defined intervention, which is difficult to meet when comparing infections with different variants that circulated during different periods of the pandemic.

Most importantly in our view, the fact that variants came in “waves” means there is a very strong association (effectively structural confounding) between time since most recent prior infection and variant of most recent prior infection, as illustrated in Figure 2. Therefore, in the BA.4/5 wave in panel (c), for example, we can only estimate the impact of last Delta infections 180-450 days ago, whereas we can only estimate the impact of last Alpha infections 420-660 days ago. Whilst this is structural confounding, biologically it is meaningless to try to estimate the impact of shorter or longer times since a last Delta infection, since the Delta variant was not circulating or had not emerged at these times, respectively. Figure 2 demonstrates that any matching approach on the type of last infection would lead to the vast majority of participants being excluded from analyses, risking substantial lack of generalisability (a point noted in Dagan et al, which excluded 33% of vaccinated participants due to lack of matching).

However, were we to even try to match the small numbers of participants with similar times from prior infections with different variants (e.g. in Figure 2 for new BA.4/5 infections those previously infected 180-300 days ago with Delta, BA.1 or BA.2 variants), the problem is that the calendar times that these matched participants would be at risk in would be different – that is, we cannot match by time since previous infection and compare variants on the same calendar dates. We therefore could not see a practical way to apply a target trial approach to assess risk of reinfection over time from previous infection with different variants, although would of course be very happy to discuss with the Reviewer potential solutions that we may have missed. We have also expanded on this limitation in brief in the Discussion.

2. Hazard ratios:

The authors study an ambitious number of questions, including Ct values, waning protection, and protection against variants. However, there seems to be strong assumptions around variables of interest. Hazard ratios were calculated per 60 days after their most recent infection by a variant, though if the goal is to study hazard ratios over time (as it’s presented in Figure 2), it is unclear why time-varying HR were not used. Hernán provides a discussion of

some of these issues (cited below).

Hernán MA. The Hazards of Hazard Ratios. *Epidemiology* (2010);21(1):13–5.

Response: As mentioned above and in the original Methods, and added to Results, we used calendar date as the underlying time scale and the date of start of each wave as time 0, and then assessed the impact of time since most recent previous infection using time-updated covariates, corresponding to the time-varying HR the reviewer recommends. We grouped time from most recent infection because days from most recent infection and calendar date would be collinear within an individual, thus reducing period-cohort (identifiability) effects, categorised as 120-180, 180-240, 240-300, 300-360, 360-420, 420-480, 480-540, 540-600, 600-660, 660-720, >720 days. The risk was presented versus a reference category of 120-180 days from an infection in the wave starting ~6 months before the current wave (Delta for BA.1 and BA.2 waves, BA.1 for BA.4/5 waves, and BA.2 for BQ.1/CH.1.1/XBB.1.5 waves). We modelled its effect both categorically (Figure S6, reproduced below) and to increase power and provide more interpretable effects as a trend across the categorical time-updated variable; the HRs per 1 unit increase in the categorical variable corresponding to 60 day blocks can therefore be considered as an average risk per 60 days from the most recent previous infection across the epochs.

If the reviewer is referring to the HR for the effect of time-varying time since previous infection varying across the epoch considered in each wave, then we would first note that the BA.1 and BA.2 waves are only ~2 months long, and the BA.4/5 and BQ.1/CH.1/XBB waves were only ~4 months, severely limiting power to detect any such effects even if they were present. The effects presented are average effects over these relatively short timescales.

Figure S6

3. Testing:

Testing is one of the most important factors in understanding bias as it relates to detection of infections. There should be more discussion of biases because of testing, and how authors accounted for this bias in the study design. The authors state that:

"From 28 February 2020 to 13 March 2023, 245,895 participants ≥ 18 y were infected with SARS-CoV-2 based on a positive swab test in the study, national testing programmes or self-reported at any time up to their final study assessment (see Methods; self-report only in 43,246 (17.6%))."

However, the percentage of self-reports (17.6%) is for first infection only, rising to 28.2% for their first reinfection. The CIS study also stopped in-person visits 14 July 2022, and collected opt-in questionnaires and testing (mailed). Does this mean that routine testing only occurred between enrollment and 14 July 2022? Would tests that were mailed in count as CIS or self-report? What proportion of participants mailed in their tests, and did they do it on-schedule? Even prior to 14 July 2022, did study workers successfully visit all participants on-schedule, and if not, what was the missingness?

Since enrollment continued until 31 January 2022, when a substantial proportion of the population had already been infected, what assumptions were made regarding previous infections? How did the study design account for some of these biases? These questions are all critical to understand potential biases in data collection. Depending on the answers to these questions, it may also make analyses for most of BA.4/5 wave, and all of BQ.1/CH.1.1/XBB.1.5 wave distinct, and potentially not comparable with other waves.

Response: The design of the survey was described in some detail in the original Methods, however we have taken this opportunity to add some brief detail to the start of the Results section in the manuscript to address the reviewer's concerns, as well as amending the Methods to for clarification.

In terms of the reviewer's specific points, study testing occurred between enrolment and 13 March 2023 for all participants after which the data collection was officially paused. After 31 July 2022, study worker visits were discontinued and participants returned test kits by post or courier, which was counted as study testing rather than self-report, since the study changed the mode of data collection for all participants but still tested all the swabs centrally. There was minimal impact on positivity based on analyses of the cross over period
<https://www.ons.gov.uk/peoplepopulationandcommunity/healthandsocialcare/condition sanddiseases/methodologies/coronaviruscovid19infectionsurveyqualityreportaugust2022>;
reference added to Results and Methods). Self-report only included tests performed outside of the survey and reported by participants on questionnaires (clarified in Methods).

Although enrolment continued until January 2022, 65% of participants were recruited by December 2020, the start of the Alpha wave, and 84% before May 2021, the start of the Delta wave (added to Methods). We used all information available from before enrolment (from linked national testing programmes and self-reported tests) to identify prior infections (clarified in Methods) – although of course these are under-ascertained in the original wave, as for every study of reinfection risk. We included participants self-report of thinking that they had COVID-19 in the first wave (and later) regardless of test confirmation in the new sensitivity analysis, without any change in results.

The survey design was to assess participants for swab tests on an approximately monthly basis. In practice this was implemented through +/- 14 day windows around assessments, targeting 28-42 days between visits, in order to achieve overall targets for swabs samples per month (as per the approved protocol, available on <https://www.ndm.ox.ac.uk/covid-19/covid-19-infection-survey/protocol-and-information-sheets>). It is true that if participants were severely ill or in hospital, there might be missed assessments, but as this was a community-based study, the majority of infections were mild. Further, by design, study workers would continue to visit the participants even if the participants had symptomatic infections (up to 31 July 2022) and after this swab samples could be returned by post or courier, even if participants had symptomatic infections, so there was no reason for infections to be missed if participants were symptomatic.

To provide further information about study assessments, we have added the distribution of the intervals between study assessments below and as Figure S12. The overall median duration between study assessments was 28 days (IQR 26-35). Most of the intervals between assessments are <45 days. If an assessment was shifted, the following assessment was also shifted to avoid swabbing participants again at very short (and variable) notice, making it difficult to assess "missed" tests except through comparison of these intervals between assessments. Therefore, whilst a small number of infections may have been missed because of missed assessments, which we have added as a limitation in the manuscript, the survey still assessed participants regularly reducing the impact from missed assessments, particularly compared with symptom/case contact driven testing in national testing programmes. The median duration between study worker visits before 31 July 2022 was 28 days (IQR 25-32) (including skew from weekly testing during the first month after enrolment), and between participants returning test kits by post or courier was 35 days (IQR 31-40). The intervals were slightly longer after 11 July 2022 for remote data collection, reflecting increased pressure to manage costs and ensure that the total number of tests per month remained on target, but participants still performed and posted the tests on schedule. We have added and discussed this as a limitation.

A total of 467295 participants contributed to the survey from study worker home visits before 31 July 2022. Among those who remained in the survey in May 2022 and were therefore invited to move to remote data collection(N=338463), 324903 participants (96%) chose to continue. 327479 participants contributed to remote data collection. There were very small differences in demographics among the initial cohort and those who continued to participate in remote data collection vs study worker visits, as shown below and added as Table S5. Therefore, the majority of participants remained in the survey after it moved to remote data collection in July 2022, and the study assessments were on relatively similar schedules, making the BA.4/5 and BQ.1/CH.1.1/XBB.1.5 waves comparable to previous waves. Further we would note that we do not make any direct comparisons across waves, but compare only time since previous infection with different variants within each wave.

	Study worker home visits (before 31 July 2022) N=467,376	Remote data collection (vast majority after 11 July 2022*) N=319996
Age		
Median	54	56
IQR	39, 67	42, 68
Sex		
Female	250233 (54%)	174292 (54%)
Male	217143 (46%)	145704 (46%)
Ethnicity		
Non-white	32811 (7%)	20612 (6%)
White	434565 (93%)	299384 (94%)
Reporting working in healthcare		

No	446558 (96%)	304659 (95%)
Yes	20818 (4%)	15337 (5%)
Reporting having a long-term health condition		
No	342908 (73%)	235145 (73%)
Yes	124468 (27%)	4851 (27%)

4. Ct values:

The Ct analysis provides interesting information, but their discussion should be done cautiously and include caveats of issues raised around interpreting Ct values (more below). Because testing was not frequent (mostly monthly) and not always routine, there may be a number of reasons why Ct values increased with reinfections, such as shortened duration of reinfections, less vigilant self-testing among those with previous infections, etc.

It may also be advisable to filter the Ct analysis to only include specimens analyzed in the same laboratory and using the same method. There was also a large fraction of specimens with missing Ct values, and there should be a greater discussion of how the missing values occurred, and whether they were systematic in any way as to bias results.

According to a joint statement by the Infectious Diseases Society of America and Association for Molecular Pathology,

“Ct values may not be comparable for individual patients tested sequentially with the same method and are definitely not directly comparable across different real-time PCR tests or testing laboratories.”

IDSA and AMP joint statement on the use of SARS-CoV-2 PCR cycle threshold (Ct) values for clinical decision-making. March 12, 2021. <https://www.idsociety.org/globalassets/idsa/public-health/covid-19/idsa-amp-statement.pdf>

Response: The Ct values used and analysed in our study were all from the same TaqPath assay and were analysed at one of the UK’s three national Lighthouse Laboratories using an identical methodology. For participants in England and Wales, we also included positive swab test results (SARS-CoV-2 PCR and lateral flow device tests) that could be linked from national clinical/hospital-based testing and community testing programmes. A substantial proportion of these additional tests were performed at the same Lighthouse Laboratories using the same test as used in CIS: we only used Ct values from these laboratories in our analyses and did not use other laboratories in the national testing programme. Therefore, we ensured that the specimens were analysed in the same settings with the same methods. We have amended the text to try to make this clearer throughout.

In Table 1 and results, we mentioned that missing Ct values were largely from infections identified from national testing only and not using this test (45.5% of the missing Ct) and self-report only (52.1% of the missing Ct). We did not analyse Ct data for infections identified through these methods for precisely the reasons the reviewer is concerned about. To examine the robustness of our results, we fitted a separate linear regression

model only including infections with Ct values available from CIS positive swab results to examine the association between Ct values and reinfection. Results remained very similar. We have added this as Figure S3.

Minor comments:

1. The authors use strong causal language, and more measured language is recommended.

For example,

“Protection” -> “Estimated protection”

“Risk against” -> “Estimated risk”

“Reinfection” -> “Detected/confirmed reinfections”

Response: We have modified our language to be less causal throughout the manuscript as suggested. We do agree with the reviewer that the models we use can only estimate associations (eg comparing the risk of reinfection in the kind of people who are 300-360 days after an Alpha infection vs the kind of people who are 360-420 days after an Alpha infection) but, as always, the challenge is to write concisely and understandably whilst doing so (amending the language has added a reasonably number of words). We would be happy to amend any other specific instances the reviewer has concerns about.

2. The background prevalence model seems to be more conservative and account for additional sources of variation. Is there a reason why it isn't the primary model?

Response: See response to reviewer 1 above – we have switched these two analyses as suggested.

3. In the introduction, the authors claim that routine testing programmes were a limitation of prior studies, but routine testing is known to reduce detection bias.

Response: As we stated in the original Introduction, routine (national) testing programmes are subject to bias from test seeking behaviours and access to testing. For example, people with symptomatic infection, people who are highly motivated to be tested, including based on demographics or impact on work attendance, and people with easier access to tests (e.g. less deprived) may be more likely to seek testing. This would generate a selection bias in the sample tested through national testing programmes. In contrast, the COVID-19 Infection Survey undertook regular longitudinal testing of randomly selected participants. The study design was to assess participants on an approximately monthly basis no matter whether the participant had symptomatic infection or not. This design could reduce selection bias and could be an advantage over other studies. We have clarified in the Introduction that we mean national testing programmes and removed the word routine since this seems to have led to confusion.

4. Page 12:

“Although the study design was to assess participants every 28-42 days regardless of symptomatology, and most intervals between assessments were <45 days”.
Should this be >45 days?

Response: This should be <45 days, as shown in the new Figure S12.

Reviewer #3 (Remarks to the Author):

This manuscript presents a sweeping, yet in-depth look at reinfections in the UK during the various Omicron waves through early 2023. It finds that reinfections are common, with Kaplan-Meier estimates over 50% of those infected with earlier variants being reinfected during the Omicron waves. Other key findings are that infection with a recent variant seems to offer significant protection against reinfection for a time, though infection with earlier variants offer much less protection. Likewise, vaccination offers protection against reinfection that wanes over time. Reinfections appeared to generally be less significant disease, with higher Ct values and fewer symptoms.

While the study design precludes any estimate of the absolute reduction in infection risk from a prior infection, it does provide an important look at how that protection changes over time and changes in variant, as well as the impact of vaccination and patient characteristics on risk.

The methods are sound and multiple sensitivity looks at the primary analysis (tables S3-4, figures 2, S7-9) give confidence in the results.

The in-depth look at the seven patients with five infections feels like a case series report inserted into the larger study. I believe these data are the least important (or at least most tangential to the main thrust of the paper), and yet it is presented in the first paragraph of the results. It could be removed or moved later in the results.

Response: We have moved the paragraph about the seven participants with five infections to the end of the first section of the Results as suggested by the reviewer.

Symptom data are incomplete, with symptoms identified through national testing classified as “other” even though the specific symptoms are unknown. These symptoms could be either classic or other, so I believe that it’s inappropriate to classify them as other. These unknown symptoms should be treated as a separate category or excluded, or symptoms could be treated as yes/no for the primary analysis.

Response: For the symptom analyses, we first fitted a logistic regression model with the outcome as ‘any symptoms reported vs no symptom reported’ overall. Then, we fitted an additional multinomial logistic regression model with outcome as ‘no symptom, classic symptom, and other symptom’ only among those who had study questionnaires reporting presence or absence of specific symptoms (i.e. excluding those reporting or not reporting symptoms in national testing data only). The results remain very similar to our original analyses (Table 2). For all other analyses using symptom as a covariate, we combined symptoms together and modelled as ‘any reported symptom vs no reported symptom’.

The differing definitions of reinfection when treating it as an exposure or an outcome adds complexity and makes the methods harder to follow and the results less generalizable to other studies with less-complex definitions. The logic of each definition seems sound, but I wonder if a single, simpler definition for both wouldn't be more appropriate.

Response: We have clarified in the Methods text that we defined infection episodes using a single definition, and these form the basis for all analyses – reinfections were not ***defined*** differently when treated as an exposure or outcome; analysis of reinfection risk as an outcome restricted the time period to improve specificity of our variant assignment (see below).

We do accept that our definition is complex, but the main challenge is the long duration of PCR positivity at high Ct values in some participants, and the short time between reinfections with different variants (confirmed by sequencing) in others, meaning any definition that does not account for both time and likely viral load intrinsically favours one over the other. In national testing programmes where testing is predominantly driven by symptoms or contact with a symptomatic case, that is there is a high pre-test probability that a positive test is a new infection, it is much more straightforward to use simple time-based definitions. We have clarified this subtlety in the Methods.

Similarly, the authors use a stricter definition when assigning variants as an outcome than as an exposure. How many events were excluded by this stricter definition, and did the rate of exclusion vary by variant? This information could help inform on how much potential there is for misclassification bias.

Response: Every infection episode was assigned to an infection variant (Table 1); as described in the Methods, we first used whole genome sequencing if available, otherwise as Pre-Alpha/Delta/Omicron BA.2-compatible if the S-gene was detected (with N/ORF1ab/both), or as Alpha/Omicron BA.1/Omicron BA.4/5-compatible if positive at least once for ORF1ab+N (but not for the S gene, S-gene target failure, SGTF). For those without sequencing data and with only one gene-positive (N-only/ORF1ab-only) or no gene positivity/Ct data available, we assigned them to each variant type based on the dominant circulating variant in each surveillance week (>50% of positive tests in the study) to define exposure (see figure below). This rule was to ensure that each identified infection was assigned to a variant, thus correctly capturing prior infection status.

The main challenge is the fact that, by virtue of the independent sampling by design, a substantial minority of infections were identified at high Ct values meaning that the amount of virus was too low for sequencing or even S-gene positivity, with only a single gene identified, meaning that these episodes were assigned to variants based on dates (as described in the Methods).

However, when we investigated new infections as the outcome we used calendar date as the underlying risk scale so that we could flexibly model the substantial changes in risk with each wave. This meant that we could start these models time zero when we could be more confident that, even if the infections we identified had high Ct/low VL, they were very likely to be truly the specific variant. So for example, a single gene positive infection would have been assigned to Delta up to and including 12 December 2021, but to BA.1 from 13 December 2021; but our model for new BA.1 infections started from 27 December 2021 as time zero, thus excluding “putative” BA.1 infections from 13 to 26 December 2021 inclusive. During the time periods in which we analysed risk of reinfection therefore, a given variant accounted for $\geq 85\%$ of infections (as shown in the figure below), ie only a

single variant dominated in the UK. We have reordered the Methods to make it clear that this exclusion was in order to reduce misclassification bias for the outcome in the risk of infection models only.

For information, the number of excluded infections from the outcome models for BA.1, BA.2, BA.4/5 and BQ.1/CH.1/XBB infections was:

Delta switching to BA.1 (between 2021-12-13 to 2021-12-26): 1014

(3504 BA.1 infections included from 2021-12-27 to 2022-02-06)

BA.1 switching to BA.2 (between 2022-02-07 to 2022-03-13): 2300

(5644 BA.2 infections included from 2022-03-14 to 2022-05-22)

BA.2 switching to BA.4/5 (between 2022-05-23 to 2022-06-26): 2522

(15079 BA.4/5 infections included from 2022-06-27 to 2022-11-06)

The switch from BA.4/5 to BQ.1/CH.1/XBB could not be defined by S-gene positivity, as a mixture of different sub-lineages was identified by sequencing, so no infections were excluded from this epoch which ran from 2022-11-07 to 2023-03-13.

REVIEWER COMMENTS

Reviewer #2 (Remarks to the Author):

Thank you for your thorough responses. I have several followup comments that build on my prior comments.

Evolution vs waning:

Lines 223-231: "There were variable effects of time since previous infection on risk of reinfection, suggesting that, rather than solely being driven by waning over time, changes in variant were a major driver of reinfection risk in most waves. In particular, consistently across all Omicron waves, estimated protection against confirmed reinfections decreased over time from the most recent identified infection if this was the previous (thick solid lines) or penultimate (thin solid lines) variant (generally reflecting the most recent prior infection being within the preceding year), but did not change or even slightly increased over time if this most recent prior identified infection was with an even earlier variant (dashed lines) (generally more than a year previously)"

The authors make strong claims about the "drivers" of new waves (evolution vs waning). However, I'm not certain there is enough evidence to substantiate this claim.

The methods the authors use to test for significance are not the most appropriate, but even if more appropriate methods are used, non-significance cannot substantiate the conclusions the authors draw from the findings. Figure 2 seems to show decreasing HRs for reinfection by BA.4/5 and BQ.1/CH.1.1/XBB.1.5 variants when most recent infection was Pre-Alpha, Alpha, and Delta. It makes sense that p values would increase over time, since the sample sizes are smaller and smaller with each wave (as more individuals gain more recent infections). The exposures are also discretized time series data, and there are more appropriate methods for significance testing (though this reviewer isn't necessarily recommending to explicitly test for significance using these methods). Rather, to further illustrate the limitations of the current manner in which the authors test for significance. Despite non-significant p-values, the trends of decreasing HRs as time from most recent previous infection increases seem quite consistent. When results are not statistically significant, it cannot be assumed that there is no effect (i.e. did not change).

Testing:

The figures about testing are very helpful. My prior comment was regarding the consistency of testing over the course of the study. While aggregate figures are helpful, it is important to know if or how testing is different between comparison groups (variant of most recent infection) and analysis (waves) to ensure groups are comparable within each analysis.

Ct values:

As Ct values are not directly comparable between laboratories, Ct values should be split by laboratory to ensure consistency of results between the aggregate and laboratory-level analysis. Analyses should also adjust for laboratory.

Reviewer #3 (Remarks to the Author):

Thank you for your additional work in modifying the manuscript in response to reviewers' comments. I believe that you have adequately addressed the concerns that were raised. I have only a few minor comments on the revised manuscript.

Line 193: "Other symptoms were not for a first reinfection.." - Other symptoms were not what?

Line 332: "estimated the effect of time since most infection" - since most recent infection?

Lines 450-451: "but confirmed reinfections are generally less severe than the first identified infections." In the previous paragraph the authors state that they can't assess disease severity, so they can't support this statement. I think this should be removed from their conclusions.

REVIEWER COMMENTS

Reviewer #2 (Remarks to the Author):

Thank you for your thorough responses. I have several followup comments that build on my prior comments.

Evolution vs waning:

Lines 223-231: "There were variable effects of time since previous infection on risk of reinfection, suggesting that, rather than solely being driven by waning over time, changes in variant were a major driver of reinfection risk in most waves. In particular, consistently across all Omicron waves, estimated protection against confirmed reinfections decreased over time from the most recent identified infection if this was the previous (thick solid lines) or penultimate (thin solid lines) variant (generally reflecting the most recent prior infection being within the preceding year), but did not change or even slightly increased over time if this most recent prior identified infection was with an even earlier variant (dashed lines) (generally more than a year previously)"

The authors make strong claims about the "drivers" of new waves (evolution vs waning). However, I'm not certain there is enough evidence to substantiate this claim.

The methods the authors use to test for significance are not the most appropriate, but even if more appropriate methods are used, non-significance cannot substantiate the conclusions the authors draw from the findings. Figure 2 seems to show decreasing HRs for reinfection by BA.4/5 and BQ.1/CH.1.1/XBB.1.5 variants when most recent infection was Pre-Alpha, Alpha, and Delta. It makes sense that p values would increase over time, since the sample sizes are smaller and smaller with each wave (as more individuals gain more recent infections). The exposures are also discretized time series data, and there are more appropriate methods for significance testing (though this reviewer isn't necessarily recommending to explicitly test for significance using these methods). Rather, to further illustrate the limitations of the current manner in which the authors test for significance. Despite non-significant p-values, the trends of decreasing HRs as time from most recent previous infection increases seem quite consistent. When results are not statistically significant, it cannot be assumed that there is no effect (i.e. did not change).

Response: We take the reviewer's point about the most appropriate tests comparing the different associations with time from most recent infection according to variant of the most recent infection being heterogeneity tests and have now added these tests as Supplementary Tables 3B-E (example for new Omicron BQ.1/CH.1.1/XBB.1.5 Figure 2d reproduced below).

	HR (95% CI)	Pre-alpha	Alpha	Delta	BA.1	BA.2	BA.4/5
Pre-alpha	0.88 (0.63-1.22)		P=0.9	P=0.9	P=0.6	P=0.08	P=0.006
Alpha	0.85 (0.76-0.95)			P=0.5	P=0.1	P<0.0001	P<0.0001
Delta	0.89 (0.85-0.93)				P=0.04	P<0.0001	P<0.0001
BA.1	0.96 (0.91-1.01)					P<0.0001	P<0.0001
BA.2	1.19 (1.14-1.25)						P=0.003
BA.4/5	1.46 (1.30-1.64)						

Note: Heterogeneity tests comparing estimated declines following most recent prior infection with different variants, see Table S3A for all model estimates.

The reason we did not include these 4 additional tables originally is because – whilst overlapping or non-overlapping 95% CI do not perfectly align with heterogeneity tests for evidence of difference, the difference in direction and tightness of 95% CI around the estimates from Figure 2, as presented in Table S3, clearly support the statements we made in the text.

However, the comparison we are making in the Results section highlighted by the reviewer is not a comparison over time within prior variant, but a comparison of these trends across variants. That is, we are basing our claims on statistical evidence of difference (based on the heterogeneity tests above), not on evidence of no difference as suggested by the reviewer above.

We have amended the main text to read “estimated protection against confirmed reinfections decreased over time from the most recent identified infection if this was the previous (thick solid lines) or penultimate (thin solid lines) variant (generally reflecting the most recent prior infection being within the preceding year), but did not change or even slightly increased over time if this most recent prior identified infection was with an even earlier variant (dashed lines) (generally more than a year previously) with evidence of statistical heterogeneity between most recent identified infection being with the previous/penultimate vs earlier variant ($p < 0.05$; Tables S3B-E).”

Further, we would note that whilst the reviewer is correct that power affects p-values, and this is particularly the case for most recent pre-Alpha infections, in fact the 95% CI around the estimates of trend for most recent infections with Alpha or Delta variant are relatively close to 1, meaning that we can exclude the possibility of substantive *waning* over time (as indicated by a hazard ratio > 1), even given the non-significant p-value and lower power to detect effects.

Testing:

The figures about testing are very helpful. My prior comment was regarding the consistency of testing over the course of the study. While aggregate figures are helpful, it is important to know if or how testing is different between comparison groups (variant of most recent infection) and analysis (waves) to ensure groups are comparable within each analysis.

Response: We interpret this comment to mean that the reviewer would like to see Figure S12, added in the previous revision and comparing time between assessments before and after the move to remote data collection, split instead by “wave”. We have added this as panel c in Figure S12 and briefly referred to it in the Methods text (noting that the earliest waves had more of the early weekly assessments by design), accompanied by the median (IQR) days between assessments in each wave. We can see that the distributions were generally similar across waves, so the groups were comparable within our analyses.

Ct values:

As Ct values are not directly comparable between laboratories, Ct values should be split by laboratory to ensure consistency of results between the aggregate and laboratory-level analysis. Analyses should also adjust for laboratory.

Response: We do not have the laboratory for tests done using the TaqPath assay outside of CIS (i.e. in national testing programmes), only that the tests were done with this assay; therefore we cannot adjust for laboratory. We have added this to the Methods (that only the Ct values and not the laboratory were available for tests done outside CIS) and explicitly noted our inability to adjust for

laboratory in the Discussion. We have already included analyses of Ct values only from CIS tests, i.e. excluding all Ct values from the same assay but conducted in other laboratories as part of national testing programmes, in the previous revision. Within CIS, the Milton Keynes laboratory was only used up to 9 Feb 2021, and accounted for <5% of all positive tests within CIS, and only during the Alpha/pre-Alpha phases of the pandemic. During August 2020 to February 2021 when both Glasgow and Milton Keynes laboratories were running in parallel, considerable work was conducted by the Office for National Statistics to compare Ct values from the two laboratories, accounting for the stage of the epidemic (since these Ct values were reported in the weekly public bulletins) and there was no evidence of differences. Given this, we have not repeated the analysis of CIS Ct values only adjusting for laboratory. We have added this to the Methods.

Reviewer #3 (Remarks to the Author):

Thank you for your additional work in modifying the manuscript in response to reviewers' comments. I believe that you have adequately addressed the concerns that were raised. I have only a few minor comments on the revised manuscript.

Line 193: "Other symptoms were not for a first reinfection.." - Other symptoms were not what?

Response: We have corrected this as: "Other symptoms were not significantly different for a first reinfection than a first infection".

Line 332: "estimated the effect of time since most infection" - since most recent infection?

Response: We have corrected this as: "estimated the effect of time since most recent infection".

Lines 450-451: "but confirmed reinfections are generally less severe than the first identified infections." In the previous paragraph the authors state that they can't assess disease severity, so they can't support this statement. I think this should be removed from their conclusions.

Response: Here we referred to our findings that reinfections had higher Ct values (lower viral load) and fewer reported classic symptoms, while when we said we can't assess disease severity we referred to hospitalisation and death. We have added them to make the conclusion clearer: "confirmed reinfections have lower viral loads and fewer symptoms than the first identified infections".

REVIEWERS' COMMENTS

Reviewer #2 (Remarks to the Author):

Evolution vs waning

Upon rereading the main text, I realized I misunderstood a portion of their comment due to differing directionality of descriptions in the main text (protection) and Figure 2 (HR of re-infection). However, I still have the same reservations about the authors' claims.

In lines 224-226, they suggest that variable trends over time since previous infection imply changes in the variant as a major driver of reinfection risk in most waves. To substantiate this claim, the authors would likely need to calculate the reproductive number based on prior infection distribution and protection assumptions, then estimate the contribution of variants that wane vs ones that do not.

For instance, consider BA.2 reinfections. Protection wanes over time since most recent infection for Delta, while remaining stable for Alpha and Pre-Alpha. If the proportion of Alpha and Pre-Alpha previous infections was small, waning for Delta may be the main reinfection driver. If the proportion of Alpha and Pre-Alpha previous infection was higher, then it becomes more ambiguous. However, the temporal trends of protection do not decisively support their claims that changes in variant were a major driver of reinfection risk.

Testing

Ok with changes.

Ct Values

Ok with changes.

Reviewer #2 (Remarks to the Author):

Evolution vs waning

Upon rereading the main text, I realized I misunderstood a portion of their comment due to differing directionality of descriptions in the main text (protection) and Figure 2 (HR of re-infection). However, I still have the same reservations about the authors' claims.

In lines 224-226, they suggest that variable trends over time since previous infection imply changes in the variant as a major driver of reinfection risk in most waves. To substantiate this claim, the authors would likely need to calculate the reproductive number based on prior infection distribution and protection assumptions, then estimate the contribution of variants that wane vs ones that do not.

For instance, consider BA.2 reinfections. Protection wanes over time since most recent infection for Delta, while remaining stable for Alpha and Pre-Alpha. If the proportion of Alpha and Pre-Alpha previous infections was small, waning for Delta may be the main reinfection driver. If the proportion of Alpha and Pre-Alpha previous infection was higher, then it becomes more ambiguous. However, the temporal trends of protection do not decisively support their claims that changes in variant were a major driver of reinfection risk.

Response: In Figure 2, we showed the risk of reinfection by time from most previous infection and the variant of most previous infection, and we found that both waning per se and change in variants were associated with the reinfection risk. First, there was clear waning for the most recent two previous variants by time, which supported the risk from waning per se. Second, when the time from previous infection overlapped between variants, infection with the immediately preceding variant provides substantially greater protection than the one before. For example, at the same time from the most recent identified previous infection, re-infection in the BQ.1/CH.1.1/XBB.1.5 wave was lower if that most recent infection (within the last 300 days) was BA.4/5 than BA.2, and BA.2 than BA.1; similarly re-infection in the BA.4/5 wave was lower if that most recent infection (within the last 300 days) was BA.2 than BA.1, and BA.1 than Delta; and re-infection in the BA.2 wave was lower if that most recent infection (within the last 180 days) was BA.1 than Delta. Thus, both the variant of the most recent identified infection and waning of protection from the most recent identified infection if this was with the previous or penultimate variant were independently associated with reinfection risk in most waves.

We have modified our original paragraph in the results section to make this clearer, and avoided claiming that changes in variant were a major driver of reinfection risk. We have toned down and changed our conclusion to 'both viral evolution and waning immunity are independently associated with reinfection' in the abstract and discussion.